# Effects of combination therapy of a CDK4/6 and MEK inhibitor in diffuse midline glioma preclinical models

Yusuke Tomita[1,2], Gabrielle Link[3], Yi Ge[4,5], Emma R. H. Gold[4,5], Anna Racanelli[4,5], Herminio Joey Cardona[1], Megan Romero[1], Samantha Gadd[6], Jun Watanabe[1], Eita Uchida[1], Rintaro Hashizume[1,7,8,9,10], Nozomu Takata[11,12], Gonzalo Pinero[5,13], Dolores Hambardzumyan[5,13], Ivan Spasojevic[14], Guo Hu[15,16], Tammy Hennika[16,17], Daniel J. Brat[18], Adam L. Green[3]*, Oren J. Becher[1,4,5,7,8]*

1 Department of Pediatrics, Feinberg School of Medicine, Northwestern University, Chicago, Illinois, United States of America, 2 Department of Neurosurgery and Neuroendovascular Surgery, Hiroshima City Hiroshima Citizens Hospital, Hiroshima, Japan, 3 Morgan Adams Foundation Pediatric Brain Tumor Research Program, Department of Pediatrics, University of Colorado School of Medicine, Aurora, Colorado, United States of America, 4 The Jack Martin Fund Division of Pediatric Hematology-Oncology, Mount Sinai Kravis Children's Hospital, New York, New York, United States of America, 5 Department of Oncological Sciences, Mount Sinai Health System, New York, New York, United States of America, 6 Department of Pathology, Ann & Robert H. Lurie Children's Hospital of Chicago, Chicago, Illinois, United States of America, 7 Division of Hematology-Oncology and Stem Cell Transplant, Ann & Robert H. Lurie Children's Hospital of Chicago, Chicago, Illinois, United States of America, 8 Department of Biochemistry and Molecular Genetics, Feinberg School of Medicine, Northwestern University, Chicago, Illinois, United States of America, 9 Division of Pediatric Hematology-Oncology, Children's of Alabama, Birmingham, Alabama, United States of America, 10 O'Neal Comprehensive Cancer Center, University of Alabama at Birmingham, Birmingham, Alabama, United States of America, 11 Center for Vascular and Developmental Biology, Feinberg Cardiovascular and Renal Research Institute, Northwestern University, Chicago, Illinois, United States of America, 12 Simpson Querrey Institute for BioNanotechnology, Northwestern University, Chicago, Illinois, United States of America, 13 Department of Neurosurgery, Icahn School of Medicine at Mount Sinai, New York, United States of America, 14 Department of Medicine - Oncology, Duke University School of Medicine and PK/PD Core Laboratory, Duke Cancer Institute, Durham, North Carolina, United States of America, 15 Department of Molecular and Human Genetics, Baylor College of Medicine, Houston, Texas, United States of America, 16 Department of Pediatrics, Duke University Medical Center, Durham, North Carolina, United States of America, 17 Genentech, Inc., South San Francisco, California, United States of America, 18 Department of Pathology, Feinberg School of Medicine, Northwestern University, Chicago, Illinois, United States of America

* oren.becher@mssm.edu (OJB); adam.green@cuanschutz.edu (AG)

## Abstract

### Background

Diffuse midline glioma (DMG) is an incurable brain cancer without a single FDA-approved drug that prolongs survival. CDK4/6 inhibitors have been evaluated in children with DMG with limited efficacy. Since MAPK pathway activation is upstream of cell proliferation, we hypothesized that MEK inhibitors may increase the anti-tumor effects of CDK4/6 inhibitors. Here, we evaluated the efficacy of the CDK4/6 inhibitor ribociclib and the MEK inhibitor trametinib in human and murine DMG models to investigate combinational effects.

**Data availability statement:** All microarray files are available from the Gene Expression Omnibus (accession number GSE184786).

**Funding:** This work was supported by a grant to OJB titled Targeting MEK and CDK4/6 in Diffuse Intrinsic Pontine Glioma by the Andrew McDonough B+ Foundation (www.BePositive. org). OJB is also supported by the Cristian Rivera Foundation, Madox's Warriors, and the Steven Ravitch Chair in Pediatric Hematology-Oncology. The funders had no role in study design, data collection and analysis, decision to publish, or preparation of the manuscript.

**Competing interests:** The authors declare no competing financial interests.

**Abbreviations:** ADPC: average daily percent change; BrdU: bromodeoxyuridine; CDK4/6: cyclin dependent kinase 4 and 6; DAPI: 4',6-diamidino-2-phenylindole; DIPG: diffuse intrinsic pontine glioma; DMG: diffuse midline glioma; GEM: genetically engineered mouse; GFP: green fluorescent protein; GSEA: gene set enrichment analysis; H&E: Hematoxylin and eosin; LC/MS: liquid chromatography-mass spectroscopy; MAPK: mitogen-activated protein kinase; MEK: MAP kinase-Erk-kinase; Np53fl: Nestin-Tv-a;p53$^{fl/fl}$; PDGFA: platelet-derived growth factor-A; PDGFB: platelet-derived growth factor-B; PDGFRA: Platelet-derived growth factor receptor A; pH3Ser10: phospho-Histone H3 Serine-10; RB: retinoblastoma protein; RCAS: Replication-Competent Avian sarcoma leukosis virus long terminal repeat with a Splice acceptor; RTK: receptor tyrosine kinase; pS6RP: S6 ribosomal protein phosphorylation; TP53: Tumor Protein 53; Tv-a: Tumor virus A.

## Methods

We conducted *in vitro* and *in vivo* assays using DMG cell lines from human patient-derived xenografts (PDX) and genetically engineered mouse (GEM) models. *In vitro,* we assessed synergy across human DMG lines. *In vivo*, we evaluated therapeutic effects with histological examinations, survival analysis, pharmacokinetic measurements, and RNA-sequencing analysis.

## Results

*In vitro*, ribociclib and trametinib had variable synergistic effects against human DMG cell lines. *In vivo*, a five-day treatment with combination therapy in the GEM DMG model significantly decreased cell proliferation and increased apoptosis compared with the vehicle, with trametinib having mostly cytotoxic effects and ribociclib having primarily cytostatic effects. In addition, a 21-day treatment with combination therapy significantly prolonged mice survival compared with the vehicle in the GEM DMG model (median survival: 112 days vs. 71.5 days, log rank test p = 0.0195). In an orthotopic PDX model, combination therapy did not prolong mice survival compared with vehicle, ribociclib, and trametinib. LC/MS analysis showed adequate drug delivery across the blood-brain-barrier (BBB) into tumor tissue in both GEM and PDX models. Transcriptomic analysis in the GEM model suggests that combination therapy inhibited the MAPK pathway and inflammation.

## Conclusions

Combination therapy with ribociclib and trametinib significantly prolonged survival in the GEM model but not in the PDX model, highlighting the importance of testing novel therapies in diverse models.

---

## Importance of the study

DMG is an incurable brain cancer without a single FDA-approved drug that prolongs survival. CDK4/6 inhibitors have been evaluated in children with DMG with limited efficacy. As MAPK pathway activation is known to be upstream of cell proliferation, we hypothesized that combination therapy will have additive effects relative to monotherapy. In this study, we evaluated the efficacy of the CDK4/6 inhibitor ribociclib and the MEK inhibitor trametinib in GEM and PDX DMG models. *In vitro*, the inhibitor combination demonstrated variable synergy in DMG human cell-lines, but *in vivo*, it had limited efficacy in a PDX orthotopic model. In the GEM model, the combination prolonged survival compared with the vehicle and with single agents. PK studies showed similar drug levels in tumor tissue between the GEM and PDX models. Transcriptomic analysis in the GEM suggests that the drug combination significantly inhibited inflammation. These results highlight the importance of testing therapies in diverse models and suggest that this combination may be an effective therapy in patients with DMGs with PDGFRA alterations and H3.3K27M.

## Introduction

Diffuse midline glioma (DMG) accounts for 15–20% of pediatric brain tumors and is a leading cause of tumor-related deaths in children [1]. Numerous clinical trials for patients with DMG have been attempted in the past 50 years, and the only effective therapy thus far is focal radiation [2]. The failures of these trials may be partly due to the unique genetic profile of DMGs. Seminal studies have revealed that somatic gain-of-function *K27M* mutations in histones H3.1 and H3.3 account for as much as 85% of DMGs [3–6]. *H3K27M* mutation causes aberrant Polycomb Repressive Complex 2 activity, causing global loss of but also focal gain in *H3K27*me3 [7]. The mutation's effects on *H3K27* methylation collectively promote tumor growth.

One of the genes suppressed by *H3K27M* is *INK4a*, or *p16*, an endogenous inhibitor of cyclin dependent kinase 4 and 6 (CDK4/6) which regulates the phosphorylation of the retinoblastoma protein (RB) and the G1-S cell cycle transition. We and others have previously shown that the *H3.3K27M*-dependent repression of *p16* is important for *H3.3K27M*-mediated gliomagenesis, and that this may be a therapeutic target for DMG [8,9]. Based on these experimental results and observation by others [10], phase I/II studies with palbociclib and another CDK4/6 inhibitor, ribociclib, were conducted for patients with refractory/progressive CNS tumors, including DMG and newly diagnosed DMG. Both inhibitors showed feasibility but limited efficacy [11,12]. Therefore, the search for a combination therapy with CDK4/6 inhibitors is urgently needed, as DMG is a complex, heterogeneous disease, and single agents are unlikely to have sustained effects [13].

The mitogen-activated protein kinase (MAPK) is one of well-known downstream pathways activated by receptor tyrosine kinase (RTK) signaling and is a key mitogenic pathway regulating proliferation [14–16]. *PDGFRA* (platelet-derived growth factor receptor A), known as one of type III RTKs, is the predominant target of focal amplification in pediatric high-grade gliomas, including DMG [17,18]. Also, recent research using human DMG specimens has shown that the MAPK pathway was highly activated, as well as other RTK signaling pathways [19–21]. In addition, MEK1/2 (MAP kinase-Erk-kinase 1 and 2) inhibition has proven quite efficacious in pediatric low-grade gliomas, suggesting that it is feasible to safely target the MEK pathway in children with brain tumors without excessive toxicity [22].

Here, we evaluated a novel combination therapy of a CDK4/6 inhibitor and a MEK inhibitor in both human and mouse DMG models. We used human DMG cell lines and observed variable synergistic effects of ribociclib and trametinib *in vitro*. In the GEM DMG model, ribociclib and trametinib showed additive effects via promoting cytostatic and cytotoxic effects, respectively, as well as significant prolongation of survival. However, the combination therapy failed to significantly prolong the survival of DMG-bearing mice in a DMG PDX model. RNAseq analysis in the GEM model demonstrated that the combination significantly inhibited the MAPK pathway and inflammation. Thus, our study highlights the importance of evaluating novel therapies in diverse models and identifies a novel combination to evaluate in patients with DMG.

## Materials and methods

### Reagents (Drugs)

Ribociclib and trametinib, a CDK4/6 inhibitor and a MEK inhibitor, respectively, were obtained from Novartis (Basel, Switzerland) and purchased from Selleck Chem. Ribociclib was dissolved in 1% hydroxyethylcellulose, and trametinib was dissolved in 5% DMSO/95% corn oil.

### Mice

For the genetically engineered murine DMG model, *Nestin-Tv-a; p53^{fl/fl} (Np53fl)* mice were created by crossing *Nestin-Tv-a* mice with *p53^{fl/fl}* mice (*C57BL/6J* background) from Jackson Labs. Genomic structure and characteristics of *Np53fl* have been previously reported [8]. For the patient-derived xenograft model, athymic *NCr-nu/nu* mice were purchased from Charles River Laboratories.

## Generation of in vivo murine diffuse midline gliomas

The murine GEM DMGs and the derived cell-lines used in Figs 1–2 were generated at either Duke University in accordance with the Duke University Animal Care and Use Committee and Guide for the Care and Use of Laboratory Animals (Protocol #A214-13–08) or Mount Sinai in accordance with the Icahn School of Medicine Animal Care and Use Committee and Guide for the Care and Use of Laboratory Animals (Protocol #IPROTO202200000034). The GEM DMG-bearing mice in the remaining figures were generated in Northwestern on a protocol that was approved by the Northwestern University Animal Care and Use Committee (Protocol #IS00009089). *In vivo* midline gliomas were generated using the RCAS/Tv-a system as previously described [8,23,24]. DF1 cells were transfected with RCAS plasmids (RCAS-PDGFA or RCAS-PDGFB, RCAS-H3.3K27M, RCAS-Cre, and RCAS-Y). Postnatal day 3–5 (P3-P5) *Np53fl* mice were intracranially injected with $10^5$ DF1 cells in 1 μl saline, producing the various RCAS viruses as described previously [25]. Mice with injection-induced hydrocephalus were euthanized and excluded from the analysis.

## Generation of orthotopic xenografts

All *in vivo* experiments were conducted in accordance with the University of Colorado Anschutz Medical Campus Institutional Animal Care and Use Committee (Protocol #00151). Athymic *NCr-nu/nu* mice were purchased from Charles River Laboratories, which were given one week to acclimatize. When mice were 6–7 weeks old, $2x10^5$ BT-245 pluc2 GFP cells were injected intracranially into the pons region of their brains. Mice were induced with 4% and then maintained at 2% isoflurane for the duration of the procedure. Mice were provided carprofen (5 mg/kg, SQ) during, one day after, and two days after injection, and then were monitored for any health changes post-surgery. An in vivo imaging system (IVIS) was used to confirm tumor engraftment and to quantify tumor growth over time. Bioluminescence was quantified for each mouse using bioluminescent flux (photons/sec). MRI was performed on a Bruker BioSpec 9.4 Tesla MR animal scanner (Bruker Medical, Billerica, MA). MRI was performed and analyzed by the University of Colorado Anschutz Medical Campus Cancer Center Animal Imaging and Irradiation Shared Resource (RRID:SCR_021980). All *in vivo* procedures were conducted by trained personnel.

## Establishment and culture of tumor cell lines

DMG cells from genetically engineered mice (14-1214-1, 14-1206-1, 14-1206-5, 23-0104-3, 23-0509-2, and 4738) were isolated from symptomatic DMG-bearing mice (*Ntv-a; p53fl/fl*; PDGFB; H3.3K27M; Cre) and were cultured as previously described [8]. The primary human DMG cell lines SF8628 and SF7761 (*H3.3K27M* DMG) were obtained from the University of California, San Francisco (UCSF, San Francisco, CA) medical center in accord with an institutionally approved protocol. The HSJD-DIPG007 (*H3.3K27M* DMG) cell line was kindly provided by Dr. Angel Montero Carcaboso (Hospital Sant Joan de Déu, Barcelona, Spain). SU-DIPG17 (*H3.3K27M* DMG) cell lines were kindly provided by Dr. Michelle Monje (Stanford University, Stanford, CA). BT-245 (patient-derived thalamic DMG) cell lines were kindly provided by Dr. Keith Ligon (Dana-Farber Cancer Institute). These human cell lines were cultured as previously described. Short tandem repeat (STR), using the Powerplex16HS System (Promega DC2101), was obtained to confirm the identity of the cell lines. All cells were cultured in an incubator at 37°C in a humidified atmosphere containing 95% $O_2$ and 5% $CO_2$ and were mycoplasma-free at the time of testing with a Mycoplasma Detection Kit (InvivoGen). The full list of cell-lines used in this study and their genetic alteration is listed in S13 Table.

## In vitro experiments (human cells)

For determining cell proliferation effects of inhibitor treatments, human tumor cells were seeded in 96-well plates at 2,000 cells per well and cultured in the presence of ribociclib and/or trametinib for 72 hours, with quadruplicate samples for each

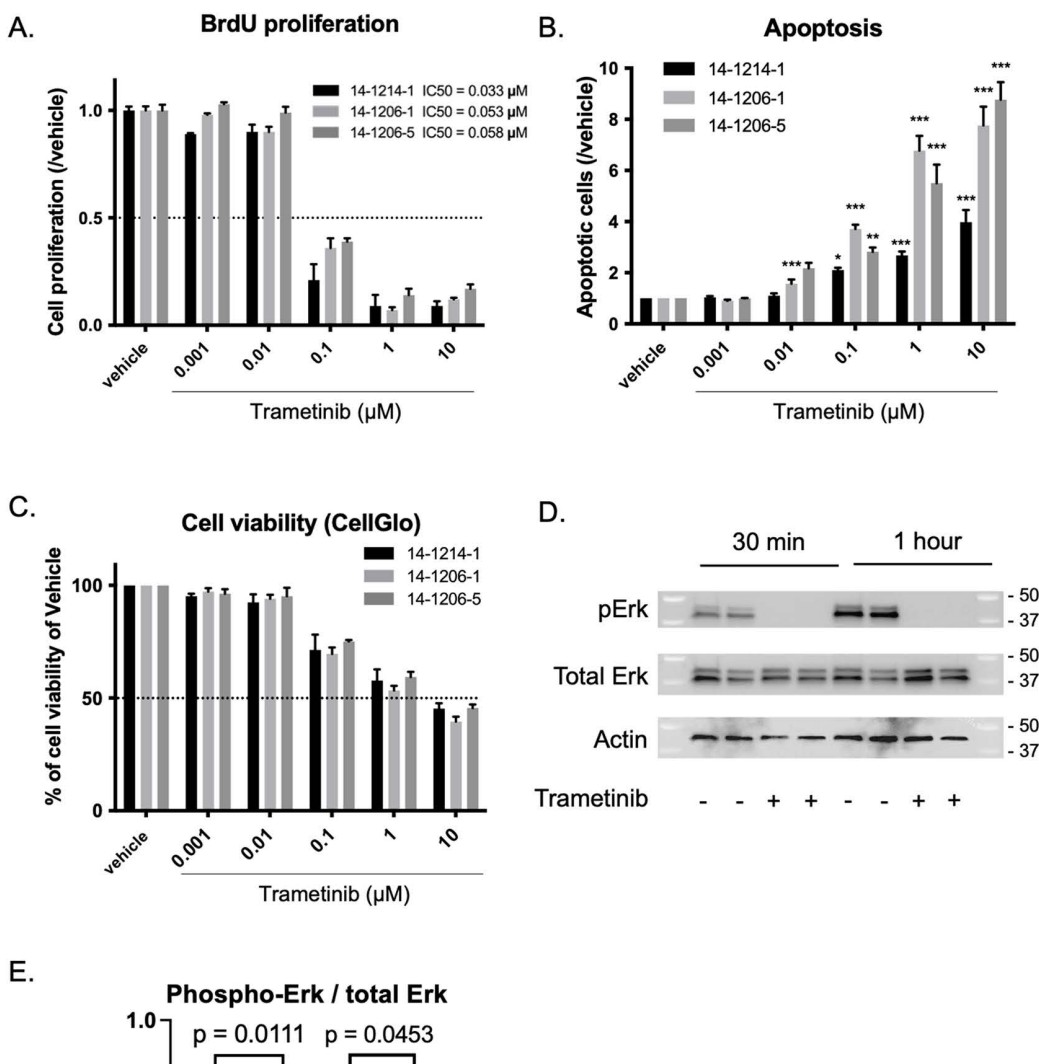

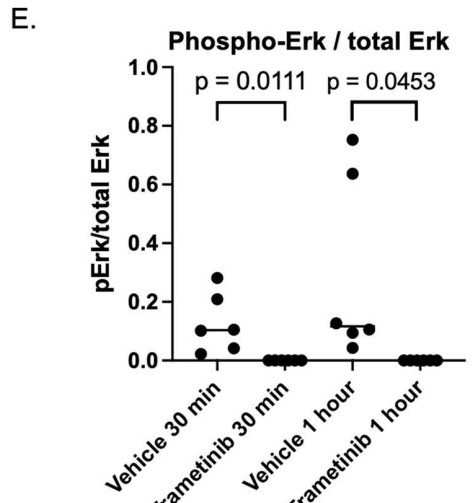

**Fig 1.** *In vitro* **effect of trametinib on PDGFB; H3.3K27M; p53-loss cells.** A. BrdU assays showed inhibition of proliferation with an IC50 of 0.033–0.058 µM. B. Apoptosis assays showed a significant increase in apoptosis compared with vehicle at 0.1 µM, 1 µM and 10 µM. Statistical significance was calculated by analysis of variance with 1-way ANOVA with Tukey's post hoc test. C. Cell-Glo assays showed a decrease in cell viability at 0.1 µM doses or higher. D-E. Western blot analysis showed inhibition of Erk phosphorylation at the protein level, with inhibition observed at 1 µM after only 30 minutes of treatment. E shows quantification of D. Mean±SEM, n=3.

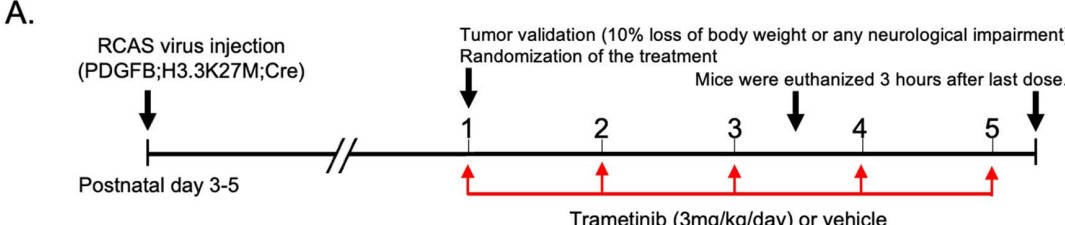

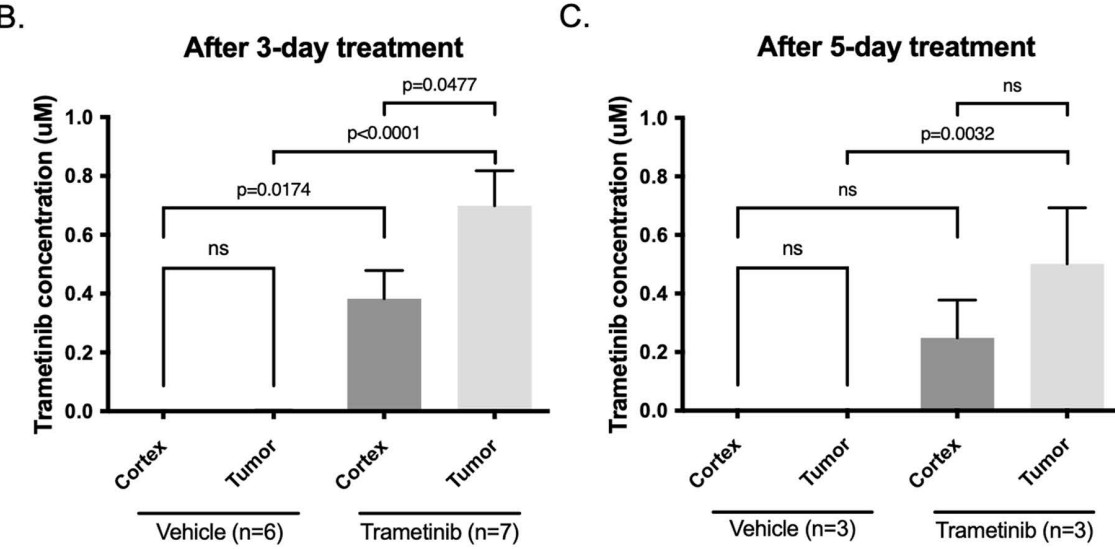

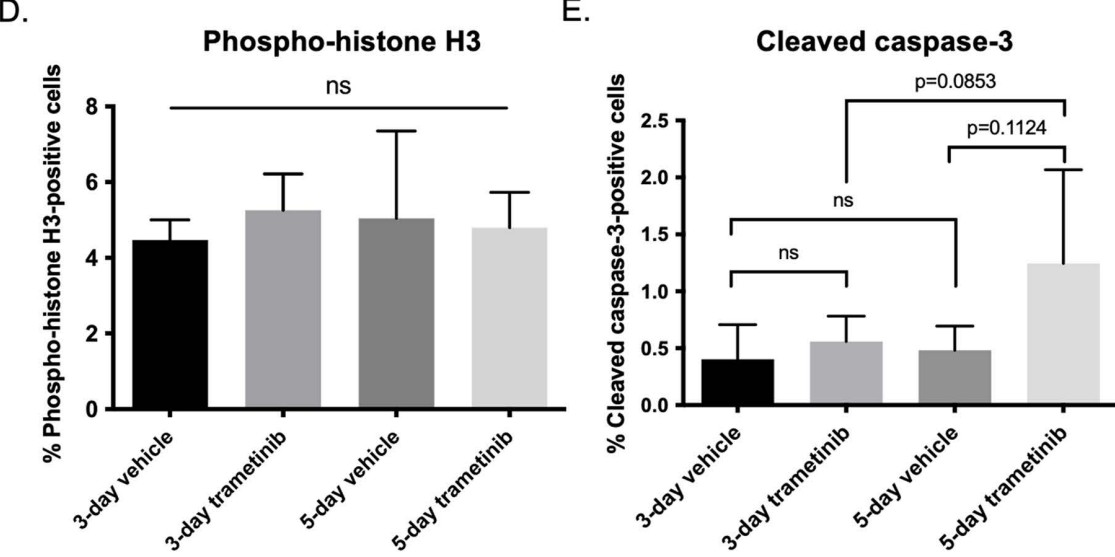

**Fig 2. *In vivo* effects of trametinib on PDGFB; H3.3K27M; p53-loss tumor-bearing mice.** A. Schematics showing the short-term *in vivo* drug treatment schedule. After tumor establishment in mice pons, mice were treated with trametinib or received vehicle for three or five days. B-C. LC/MS analysis of tissue extracts from cortex and tumor tissue after treatment for three days (B) or five days (C). Trametinib concentration in tumor tissues after three-day treatment was 0.699±0.119 µM, and that after five-day treatment was 0.501±0.111 µM. D-E. Quantification of immunohistochemistry for phospho-Histone-3 (D) and cleaved caspase-3 (E). Mitotic cells were similar between trametinib and vehicle groups, whereas apoptotic cells tended to increase by 5-day treatment with trametinib. Statistical significance was calculated by analysis of variance with 1-way ANOVA with Tukey's post hoc test.

incubation condition. Dimethylsulfoxide (DMSO) was used as a vehicle. Relative numbers of viable cells were determined using the bromodeoxyuridine (BrdU)-based cell proliferation ELISA Assay Kit (Roche) and CellTiter 96 AQueous One Solution Cell Proliferation Assay (Promega) according to the manufacturer's protocol. The 50% growth inhibition (IC50) values were calculated using nonlinear least-squares curve-fitting. To determine the synergistic interaction of ribociclib and trametinib, the data were analyzed by the highest single agent (HSA) model using Combenefit open-access software (http://sourceforge.net/projects/combenefit/) [26].

### In vitro experiments (mouse cells)

Proliferation was measured using a bromodeoxyuridine (BrdU)-based cell proliferation ELISA Assay Kit (Roche). Cells were plated in triplicates in a 96-well plate ($5\times10^4$ cells/well in 100uL) and allowed to settle for 24 hours. Cells were then treated with concentrations of trametinib (Selleck) or 0.1% DMSO as vehicle for 48 hours. The assay was processed according to manufacturer's protocol. Absorbance was read using Molecular Devices Versa Max Tunable Microplate reader at 370 and 492nM. Data analysis and statistics were performed with Prism and Excel. Cell viability was assessed by Cell-Glo assay (Promega). ApoTox-Glo Triplex Assay (Promega) was used to assess cell apoptosis. Cells were plated the same way as above for cell proliferation. Subsequently, the Cell-Glo and apoptosis assays were performed according to the manufacturer's protocol. Luminescence was read using a Turner Biosystems Modulus Microplate Reader. Data analysis and statistics were performed as above for cell proliferation.

### Western blotting

Lysates from murine PDGFB; H3.3K27M; p53-loss tumor cells were prepared using a nuclear lysis buffer containing 25 mM Tris-HCl, pH 7.6, 150 mM NaCl, 1% NP-40, 1% sodium deoxycholate, 0.1% SDS, and 1% Halt™ Protease and Phosphatase Inhibitor Cocktail (Thermo Fisher), then separated by polyacrylamide gel electrophoresis and transferred to a nitrocellulose membrane. Membranes were blocked in Tris-buffered saline with 0.1% Tween-20 and 5% non-fat milk at room temperature for one hour. Primary antibodies were prepared in 5% TBS-T and incubated overnight at 4 degrees Celsius. The following primary antibodies were used: anti-phospho-p44/42 MAPK (Erk1/2) (Cell Signaling Technology, #9101, 1:1000), p44/42 MAPK (Erk1/2) (Cell Signaling Technology #4695, 1:1000), and Actin (Cell Signaling Technology #3700, 1:1000). Secondary antibodies conjugated with horseradish peroxidase were prepared in blocking buffer and incubated at room temperature for one hour. Detection was performed with Thermo Scientific SuperSignal West Pico PLUS Chemiluminescent Substrate according to the manufacturer's protocol (Thermo Fisher). Membranes were imaged with the Bio-Rad ChemiDocTM Imaging System and quantified using Image Lab™ software.

### Short-term treatment

Tumor-bearing mice were treated with five doses of ribociclib (75 mg/kg), three or five doses of trametinib (3 mg/kg; this high dose was only used initially for the trametinib monotherapy study which has been used previously by others [27]) using PDGFB; H3.3K27M; p53-loss mice, or 0.3 mg/kg of trametinib (this low dose was used for combination studies) using PDGFA; H3.3K27M; p53-loss mice, or the vehicle (1% hydroxyethylcellulose as ribociclib vehicle or 5% DMSO/95% corn oil as trametinib vehicle) administered once daily by oral gavage. The dosage of ribociclib was determined by the previous preclinical study to evaluate the efficacy of ribociclib and binimetinib in a neuroblastoma model [28]. Drug treatments were initiated at the time point when mice lost 10% of their body weights or showed any neurological impairments for the GEM model, or at the time a significant IVIS signal was detected for the PDX model. For the GEM model, tumor volumes were not measured, as MRI and bioluminescent imaging were not used for this model. Mice were sacrificed three hours after their final treatment with $CO_2$, and brain tissues were processed and evaluated as described previously [8].

## Analysis of drug concentration in the brain

In cases of a high trametinib dose (3 mg/kg), brain tumor tissue was homogenized with 2 parts water (w/v) by rotary homogenizer. 20 µL of tissue homogenate, 10 µL of 1 µg/mL osimetrinib (internal standard), 100 µL water, and 200 µL chloroform were added to a 0.5 mL polypropylene tube and vigorously agitated in a FastPrep vortexer (Thermo-Savant) at speed 4 for 20s. After centrifugation at 16,000 g for 5 min at room temperature, the lower organic layer was transferred to a 12x75 mm glass tube, and the solvent was removed under a gentle stream of nitrogen at room temperature (~15 min). The dry residue was dissolved in a mixture of mobile phase solvents (50%A/50%B, see below), and 5 µL was injected into LC/MS/MS system. In cases of a low trametinib dose (0.3 mg/kg) and simultaneous measurement of ribociclib (75 mg/kg), 10 mg of brain tumor tissue, 10 µL of 10 ng/mL trametinib-$^{13}$C-d3 (int. std.) and 200 ng/mL palbociclib (int. std.) mixture in acetonitrile, 100 µL water, and 200 µL ethyl acetate were agitated, centrifuged, and 150 µL of the ethyl acetate layer was evaporated to dryness as described above. Into the remaining aqueous phase and the interface layer ("cake"), 20 µL of Trizol® and 200 µL chloroform was added, agitated and processed in the same manner as above, ending with dry residue containing ribociclib. Both dry residues, from ethyl acetate (containing trametinib) and chloroform/Trizol® (containing ribociclib) were reconstituted with 100 µL of 50%A/50%B each (A & B solutions described below), and 10 µL was injected as separate LC/MS/MS runs. The LC/MS/MS analysis was performed on an Agilent 1200 series LC system coupled with an Applied Biosciences/SCIEX API 5500 QTrap MS/MS spectrometer. Column: Agilent Eclipse C18 4.6x50 mm, 1.8 µm with Phenomenex, C18 4x3 mm guard at 40°C. Mobile phase solvents: A – 0.5% formic acid in water, 10 mM ammonium hydroxide, 2% acetonitrile; B – acetonitrile. Elution gradient at 1 mL/min: 0–0.2 min 5% (ribociclib method) or 30% (trametinib method)-95% B, 0.2–0.7 min 95% B, 0.7–0.8 min 95−5% (30%) B. Run time: 4 min. MS/MS transitions in positive ionization mode for trametinib, trametinib-$^{13}$C-d3, osimetrinib, and ribociclib (m/z): 616.1/491.0, 620.1/495, 500.3/385.2, and 435.2/322.2, respectively. Lower limit of quantification (LLOQ) for trametinib was and 0.81 ng/g and for ribociclib was 2.43 ng/g wet brain tumor tissue. Calibration curve samples in the appropriate range were prepared in corresponding drug-free brain tissue homogenate and analyzed alongside study samples. Quantification of study samples was performed using Analyst 1.6.2 software.

## Long-term treatment

Survival analysis was conducted in strict accordance with the recommendations in the Institutional Animal Care and Use Committee in Northwestern University and the University of Colorado Anschutz Medical Campus. Tumor-bearing mice were treated with ribociclib (75 mg/kg), trametinib (0.3 mg/kg), or vehicle (1% hydroxyethylcellulose as ribociclib vehicle or 5% DMSO/95% corn oil as trametinib vehicle) administered once daily by oral gavage for three weeks in the GEM study and once daily by oral gavage 5 days on/2 days off for three weeks in the PDX study. Drug treatments were initiated at postnatal day 50 for the GEM DMG model, and at the time when tumors were significantly visible on IVIS for orthotopic xenografts. In both tumor models, tumor-bearing mice were monitored daily for health and behavior, up to postnatal day 180. For the survival analyses, 68 mice were used for the GEM DMG model, and 32 mice for the PDX model. Mice were euthanized with $CO_2$ upon reaching humane endpoints (enlarged head, ataxia, and weight loss up to 20%, for the GEM model, or weight loss equal to or in excess of 15%, ataxia, lethargy, neurological insufficiency, and seizures, for the PDX model) within 2 hours. One mouse in the PDX study died before humane endpoints were observed. Next, their brains were extracted to snap freeze, fix in formalin, and embed with paraffin. Tumors were confirmed macroscopically or microscopically at the time of endpoint, and mice without any evidence of tumor were excluded from survival analysis. In the PDX study, MRI was performed during the second week of treatment on the four mice from each treatment group that presented with the highest BLI signal on IVIS. One mouse from the control group that would have received MRI was euthanized due to reaching humane endpoints before imaging. MRI ranges for each group were as follows: combination (14.4–57.1 mm3, n = 4), ribociclib (26–144.1 mm3, n = 4), trametinib (15.9–18.9 mm3, n = 4), and control (21.2–43.6 mm3, n = 3).

 

## Tumor grading

After euthanasia, mice brains were extracted, fixed in 20% formalin for 24–48 hours, and embedded in paraffin by NUcore Mouse Histology and Phenotyping Laboratory. Embedded tissue was cut into 5 μm sections using a Leica RM2235 microtome. Hematoxylin and eosin (H&E) staining was performed using standard protocols. GEM tumors were graded by a blinded neuropathologist (D.B.) according to the 2016 WHO classification: grade II glioma by moderately increased cellular density without mitosis; grade III glioma by increased cellular density with mitosis; and grade IV glioma by the presence of microvascular proliferation and/or pseudopalisading necrosis [29]. WHO grades III and IV are defined as high-grade.

## Immunohistochemistry

Surgically excised brains from mice were fixed with 20% formalin for 24−48 hours, embedded in paraffin, and 5 μm sections were prepared. The sections were stained with an automated processor (Discovery XT [Ventana Medical Systems, Inc.] or Bond Rx [Leica Biosystems]), as described previously [30]. Anti-H3K27M (Abcam, ab190631, 1:500), anti-phospho-Histone H3 (Cell Signaling Technology, #9701S, 1:200), anti-Ki67 (Abcam, ab16667, 1:200), anti-phospho-Rb (Cell Signaling Technology, #8180S, 1:100), anti-phospho-S6 ribosomal protein (Cell Signaling Technology, #2211S, 1:200), anti-phospho-Erk (Cell Signaling Technology, #4370S, 1:1000), anti-cleaved caspase-3 (Asp175) (Cell Signaling Technology, #9661, 1:400), anti-CD3 (Abcam, ab16669, 1:200), anti-CD4 (Cell Signaling Technology, 25229, 1:100), and anti-CD8a (Cell Signaling Technology, 98941, 1:200) were used for the staining. Samples were observed and quantified with a Lionheart XT automated microscope (BioTek, Vermont, USA) or a Nano Zoomer digital slide scanner (Hamamatsu Photonics, Hamamatsu, Japan).

## TUNEL staining

Apoptotic responses to treatment were evaluated with the DeadEnd Colorimetric TUNEL System (Promega) according to the manufacture's protocol. Slides were mounted with Vectashield with 4',6-diamidino-2-phenylindole (DAPI) (Vector Laboratories) and imaged using a Lionheart™ automated microscope (BioTek, Vermont, USA). In each group, staining without dUTP was conducted as negative control. The percentage of FL-12-dUTP-positive cells in DAPI-positive cells was assayed with Gen5 software (BioTek, Version 6.0).

## RNA-sequencing (RNA-seq) and analysis

RNA-seq and analysis were performed in collaboration with the NUSeq Core facility at Northwestern University. Total RNA was isolated from tumor tissue with the RNeasy Mini Kit (Qiagen), and 200 ng of purified RNA per sample was used for analysis. The libraries were prepared with TruSeq Stranded mRNA-seq Library Prep Kit and sequenced using the HiSeq 4000 sequencing system (Illumina). FASTQ files were aligned to the mm10 genome using RNA-STAR, and aligned reads were counted using HTSeq-count. HTSeq-count files were imported into R (https://www.r-project.org/), and differential expression analysis was performed with the DESeq2 package using default settings. Tumors treated with the vehicle were compared to those treated with ribociclib and trametinib as combination. Principal component analysis and hierarchical clustering were done in R. Normalized reads were imported into gene sequence enrichment analysis (GSEA), and standard GSEA was run with the following parameters: permutations = 1000, permutation type = gene set, enrichment statistic = weighted, gene ranking metric = signal2noise, max size = 500, min size = 15, normalization mode = meandiv. The data discussed in this publication have been deposited in NCBIs Gene Expression Omnibus (GEO, http://www.ncbi.nlm.nih.gov/geo/) and are accessible through GEO Series accession number GSE184786.

## Statistical analysis

Statistical analysis was performed using GraphPad Prism (Version 9.1.0). We examined trends in body weights over 21 days and calculated an average daily percent change (ADPC) with a 95% confidence interval using Joinpoint Regression

Program (National Cancer Institute Division of Cancer Control and Population Sciences. Joinpoint Regression Program, Version 4.9.0.0. Division of Cancer Controls & Population Sciences, National Cancer Institute. Available at: https://surveillance.cancer.gov/joinpoint/. Accessed September 9, 2020.). Survival rate was estimated using Kaplan Meier Curve analysis, and statistical significance was evaluated by a log-rank test. Continuous variables are expressed as mean ± standard error of the mean (SEM). Continuous variables were compared using Student's t test or one-way ANOVA with Tukey's post hoc test.

## Results

### Trametinib monotherapy showed efficacy *in vitro* but not *in vivo,* despite adequate delivery across BBB

We have previously shown the effectiveness of palbociclib, a CDK4/6 inhibitor similar to ribociclib, in PDGFB; H3.3K27M; p53-loss tumor models [8]. Also, a recent clinical trial has indicated the efficacy of ribociclib in newly diagnosed DMG patients [11]. For these reasons, ribociclib is expected to be effective against DMG both *in vitro* and *in vivo*. Therefore, we were first interested in determining if trametinib as monotherapy was effective against murine DMG models.

We first conducted *in vitro* assays on three independent samples of PDGFB; H3.3K27M; p53-loss DMG cells (14-1214-1, 14-1206-1, and 14-1206-5) which were each treated with various doses of trametinib (0.001 μM, 0.01 μM, 0.1 μM, 1 μM, and 10 μM) or DMSO as vehicle for 48 hours. Treatment with trametinib inhibited cell proliferations labeled with BrdU compared with the vehicle at concentrations above 0.1 μM (Fig 1A). IC50 values were 0.033 μM for 14-1214-1, 0.053 μM for 14-1206-1, and 0.058 μM for 14-1206-5, respectively. Next, we conducted an apoptosis assay by comparing caspase-3/7 activities. After treatment with trametinib, the number of apoptotic cells generally increased relative to the vehicle in all the cell lines tested, but with variable effects (Fig 1B). Lastly, we performed a Cell-Glo assay and noted decreased viability at concentrations above 0.1 μM (Fig 1C). To validate these findings, we conducted Western blotting to examine Erk and phospho-Erk expression post-treatment with 1 μM of trametinib (Fig 1D-E), and phospho-Erk expression was significantly downregulated compared with the vehicle (30 minutes treatment, $p = 0.0111$; 1 hour treatment, $p = 0.0453$, Student's *t* test). Collectively, our results suggest that treatment with trametinib is effective *in vitro* against PDGFB; H3.3K27M; p53-loss cell lines.

Next, we tested if trametinib monotherapy is effective in the GEM DMG model. We treated two cohorts of PDGFB; H3.3K27M; p53-loss DMG-bearing mice with vehicle or with trametinib for 3 days (n = 6 and 7, respectively) or for 5 days (n = 3 each) (Fig 2A). LC/MS analysis of tissue extracts revealed detectable trametinib in the cerebral cortex and tumor tissues both after three-day treatment (cerebral cortex, 0.383 ± 0.096 μM; tumor tissues, 0.699 ± 0.119 μM) and after five-day treatment (cerebral cortex, 0.249 ± 0.075 μM; tumor tissues, 0.501 ± 0.111 μM), indicating that trametinib concentrations in tumor tissues were ten times as high as IC50 shown *in vitro* (Fig 1A, 2B and 2C). Then we immunolabeled pH3Ser10 and cleaved caspase-3 to evaluate cell proliferation and apoptosis. Surprisingly, trametinib did not reduce pH3Ser10-positive cells compared with the vehicle (three-day treatment, trametinib 5.3 ± 2.5%, vehicle 4.5 ± 1.3%, $p = 0.9302$; five-day treatment, trametinib 4.8 ± 1.6%, vehicle 5.0 ± 4.0%, $p = 0.9993$, ANOVA with Tukey's post hoc test) (Fig 2D). Also, we observed a trend for trametinib to increase cleaved caspase-3-positive cells compared with vehicle after five-day treatment, but the results were not significant (three-day treatment, trametinib 0.6 ± 0.2%, vehicle 0.4 ± 0.3%, $p = 0.8837$; five-day treatment, trametinib 1.2 ± 0.8%, vehicle 0.5 ± 0.2%, $p = 0.1124$, ANOVA with Tukey's post hoc test) (Fig 2E). These results suggest that DMG-bearing mice are resistant to trametinib as monotherapy *in vivo* despite adequate penetration of the drug to the tumor tissues.

### Trametinib and ribociclib showed synergistic effects in human DMG cell lines *in vitro*

To overcome the innate resistance of DMG to trametinib, we conducted combination therapy with ribociclib and trametinib. We conducted CellTiter 96 AQueous One Solution Cell Proliferation Assay to examine if combination therapy could have synergistic effects against human DMG cell lines. SF8628, SF7761, SU-DIPG17, and HJSD-DIPG007 cells were first treated with ribociclib or trametinib as monotherapy (1 nM, 10 nM, 100 nM, 1 μM) or DMSO as vehicle for 72 hours.

Both ribociclib and trametinib inhibited cell proliferation compared with vehicle in a concentration-dependent manner in all cell lines (Fig 3A and 3B). IC50 values of ribociclib were 2.973 µM for SF8628, 0.575 µM for SF7761, 0.019 µM for SU-DIPG17, and 1.746 µM for HJSD-DIPG007, respectively. IC50 values of trametinib were 0.117 µM for SF8628, 0.065 µM for SF7761, 0.004 µM for SU-DIPG17, and 0.072 µM for HJSD-DIPG007, respectively. Next, to determine the synergistic interaction of ribociclib and trametinib, human DMG cells were treated with various concentrations of ribociclib and trametinib for 72 hours, followed by the analysis using the highest single agent (HSA) model using Combenefit open-access software (Fig 3C-3F). The Combenefit assay showed a synergistic interaction between ribociclib and trametinib, although SF8628 cells showed less synergy than other DMG cells. Based on the result, we conducted CellTiter 96 AQueous One Solution Cell Proliferation Assay again to confirm the synergistic effects of combination therapy (Fig 3G-3J). With SF8628, combination therapy significantly decreased cell viability compared with DMSO, ribociclib, and trametinib (combination vs. DMSO, p=0.0017; combination vs. ribociclib, p=0.0029; combination vs. trametinib, p=0.0264, ANOVA with Tukey's post hoc test) (Fig 3G). Among other cell lines, therapeutic effects of combination therapy varied (SF7761, combination vs. DMSO, p=0.0070; combination vs. ribociclib, p=0.1475; combination vs. trametinib, p=0.0127; SU-DIPG-17, combination vs. DMSO, p=0.0013; combination vs. ribociclib, p=0.0398; combination vs. trametinib, p=0.0642; HJSD-DIPG-007, combination vs. DMSO, p=0.0359; combination vs. ribociclib, p=0.0790; combination vs. trametinib, p=0.1155, ANOVA with Tukey's post hoc test (Fig 3H-3J). These results indicate that ribociclib and trametinib have variable levels of synergy against human DMG cell lines *in vitro*.

### Short-term treatment with ribociclib and trametinib decreased cell proliferation and increased apoptosis *in vivo*

As the combination therapy had promising effects against human DMG cell lines *in vitro*, we proceeded with an *in vivo* experiment using GEM DMG models. For combination therapy, we reduced the dose of trametinib to 0.3 mg/kg, which would deliver the IC50 concentration to the tumor tissues to minimize drug-induced toxicity. We treated four cohorts of PDGFA; H3.3K27M; p53-loss DMG-bearing mice with vehicle, ribociclib, trametinib, or ribociclib and trametinib as combination (n=5, 5, 4, 4 respectively) for five days and then collected the brain for histological and immunohistochemical analysis (Fig 4A). Histologically, all the DMG-bearing mice had tumors with high cellularity, and most tumor cells immunolabeled with anti-H3K27M antibody regardless of the treatments (Fig 4B).

To evaluate the effects of the drugs on the proliferation rate of the DMG-bearing mice, we immunolabeled tumor tissue for proliferative markers Ki67 and phospho-Histone H3 Serine-10 (pH3Ser10) (Fig 5A). Quantification of this staining shows that only the combination therapy significantly reduced both the Ki-67- and pH3Ser10-positive cells compared with the vehicle (Ki-67, p=0.019; pH3Ser10, p=0.0059, ANOVA with Tukey's post hoc test) (Fig 5B-5C). However, combination therapy did not significantly decrease cell proliferation compared with ribociclib as monotherapy (Ki-67, p=0.9365; pH3Ser10, p=0.5103, ANOVA with Tukey's post hoc test). Similarly, combination therapy did not significantly reduce Ki-67-positivity compared with that of trametinib as monotherapy (p=0.7445, l ANOVA with Tukey's post hoc test), whereas combination therapy significantly reduced the pH3Ser10-positive rate compared with that of trametinib (p=0.0032, ANOVA with Tukey's post hoc test). This data indicates that only combination therapy significantly decreased cell proliferation compared with the vehicle, and this effect was mainly attributed to ribociclib.

Apoptosis in response to drug treatment was assessed by detection of cleaved caspase-3 and TUNEL staining (Fig 5A). Immunostaining of the tumor sections with cleaved caspase-3 demonstrated that combination therapy induced significantly more apoptotic cells than the vehicle (p=0.0002, ANOVA with Tukey's post hoc test) (Fig 5D). In the TUNEL assay, combination therapy significantly increased TUNEL-positive cells compared with vehicle (p=0.0013, ANOVA with Tukey's post hoc test), ribociclib (p=0.0013, ANOVA with Tukey's post hoc test), and trametinib (p=0.0462, ANOVA with Tukey's post hoc test) (Fig 5E-5F). These results indicate that the combination of trametinib and ribociclib had a significant cytotoxic effect on DMG-bearing mice compared with that of the vehicle, ribociclib as monotherapy, or trametinib as monotherapy.

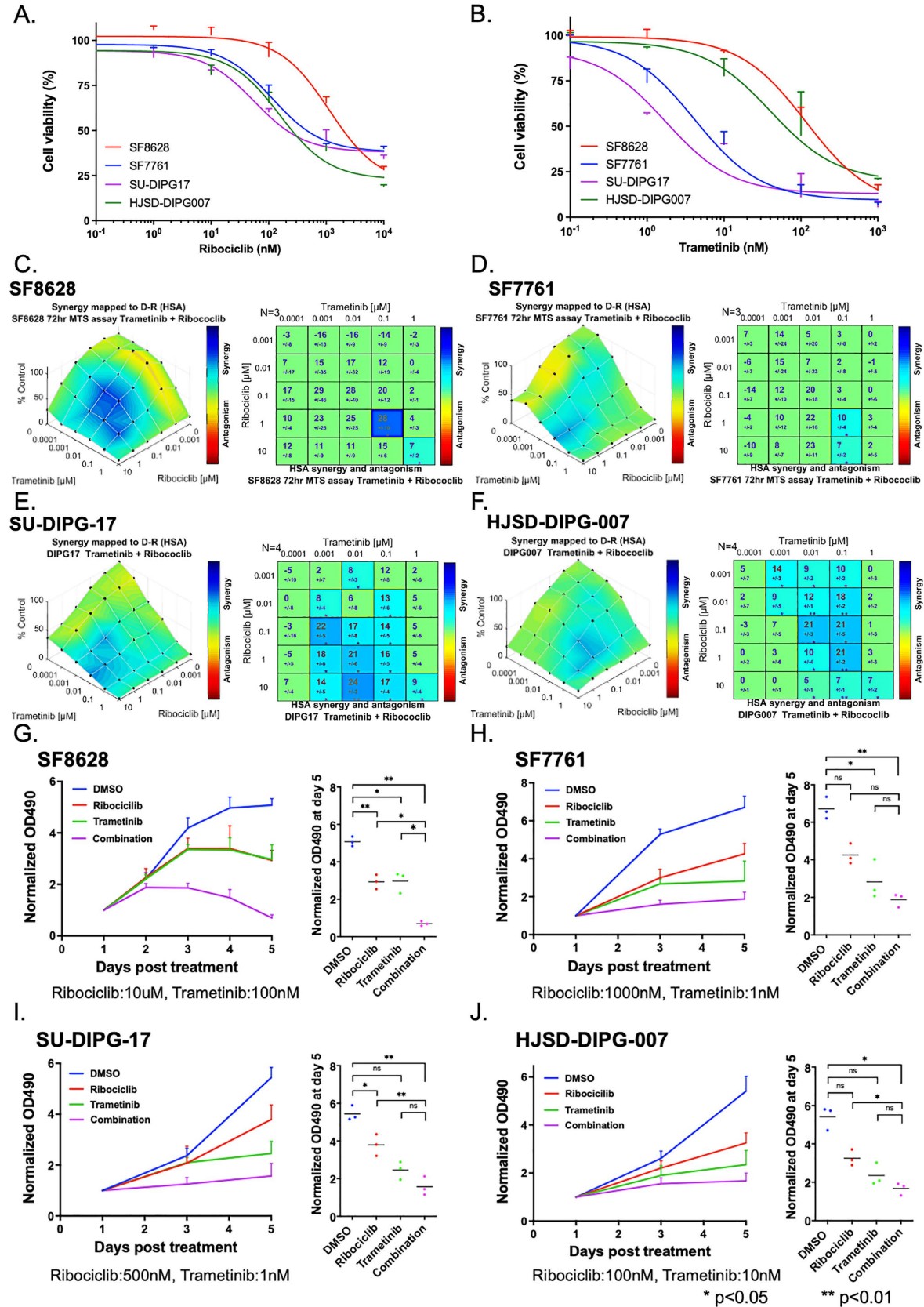

**Fig 3. *In vitro* effects of trametinib on human DMG cells.** A-B. CellTiter 96 AQueous One Solution Cell Proliferation Assay showed inhibition of proliferation from both ribociclib (A) and trametinib (B). IC50 values of ribociclib were 2.973 µM for SF8628, 0.575 µM for SF7761, 0.019 µM for SU-DIPG17, and 1.746 µM for HJSD-DIPG007, respectively. IC50 values of trametinib were 0.117 µM for SF8628, 0.065 µM for SF7761, 0.004 µM for SU-DIPG17, and 0.072 µM for HJSD-DIPG007, respectively. C-J. Synergistic effects of combination therapy were evaluated with SF8628 (C and G), SF7761 (D and H), SU-DIPG-17 (E and I), and HJSD-DIPG-007 (F and J). C-F. Combenefit analysis showed synergistic effects of ribociclib and trametinib against each cell line. G-J. CellTiter 96 AQueous One Solution Cell Proliferation Assay showed that combination therapy with ribociclib and trametinib was more effective than DMSO, ribociclib as monotherapy, and trametinib as monotherapy. Statistical significance was calculated by analysis of variance with ANOVA with Tukey's post hoc test.

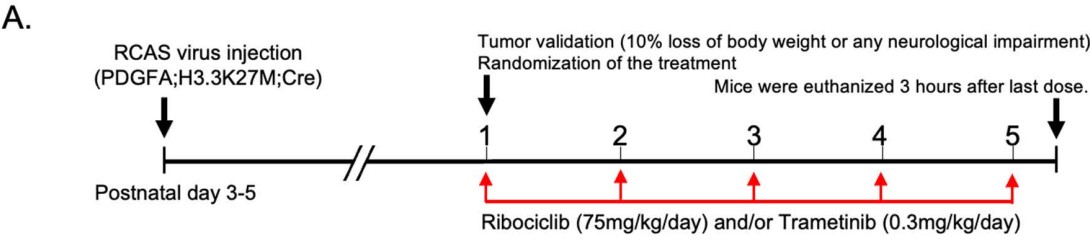

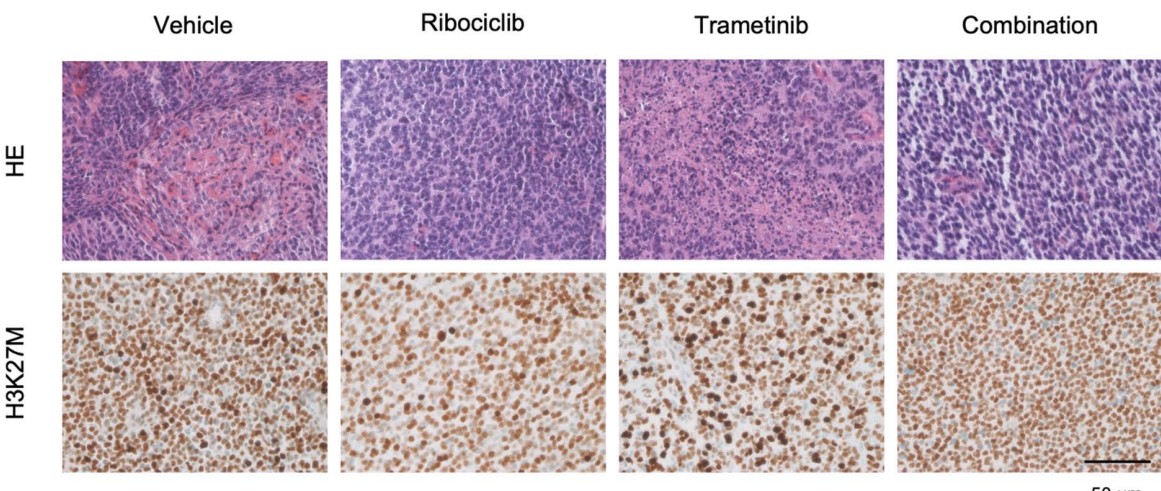

**Fig 4. Genetically engineered mice with DMG treated with ribociclib and trametinib for 5 days.** A. Schematics showing the short-term *in vivo* drug treatment schedule. After tumor establishment in mice pons, the animals were randomized into four treatment groups: vehicle, ribociclib, trametinib, and combination therapy. B. H&E staining and immunohistochemistry staining with H3K27M. All the DMG-bearing mice had glial tumors with high cellularity, and most of the cells were immunolabeled with anti-H3K27M antibody regardless of the treatments.

## Combination therapy significantly increased the limited activity observed with monotherapy *in vivo*

Next, we aimed to determine if this combination therapy inhibited the respective pathways *in vivo*: Rb phosphorylation for the CDK4/6 inhibitor, and Erk phosphorylation for the MEK inhibitor. Therefore, we performed immunohistochemistry staining of the tumor tissues with anti-pRb and anti-pErk antibodies (Fig 6A). Combination therapy, but not ribociclib as monotherapy, significantly decreased pRb positive cells compared with vehicle and with trametinib (combination vs. vehicle,

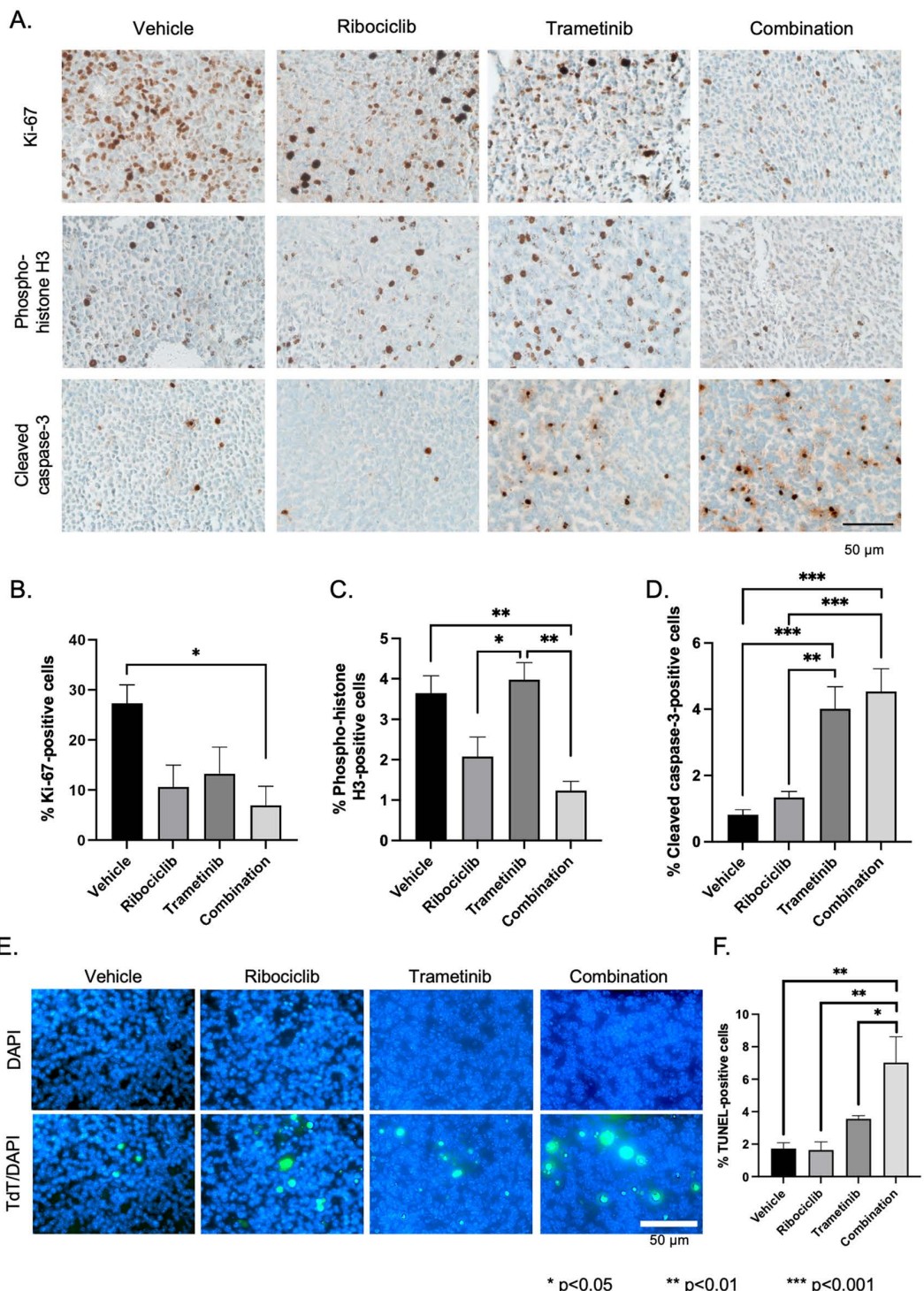

* p<0.05          ** p<0.01          *** p<0.001

**Fig 5. Effects of short-term treatment on cell proliferation and apoptosis.** A. Representative photos of Ki-67, phospho-Histone-3 (serine 10), and cleaved caspase-3 immunohistochemistry. B-D. Quantification of Ki-67 (B), pH3 (C), and cleaved caspase-3 (D). Tissues treated with ribociclib had fewer mitotic cells, whereas those treated with trametinib had more apoptotic cells. Combination therapy significantly decreased Ki-67- and phospho-Histone-3 (serine 10)-positive cells and increased cleaved caspase-3-positive cells. E. Representative photos of TUNEL staining. F. Quantification of E. Combination therapy significantly decreased TdT-positive cells. Pairwise comparison; *p<0.05, **p<0.01, ***p<0.001. Statistical significance was calculated by analysis of variance with 1-way ANOVA with Tukey's post hoc test.

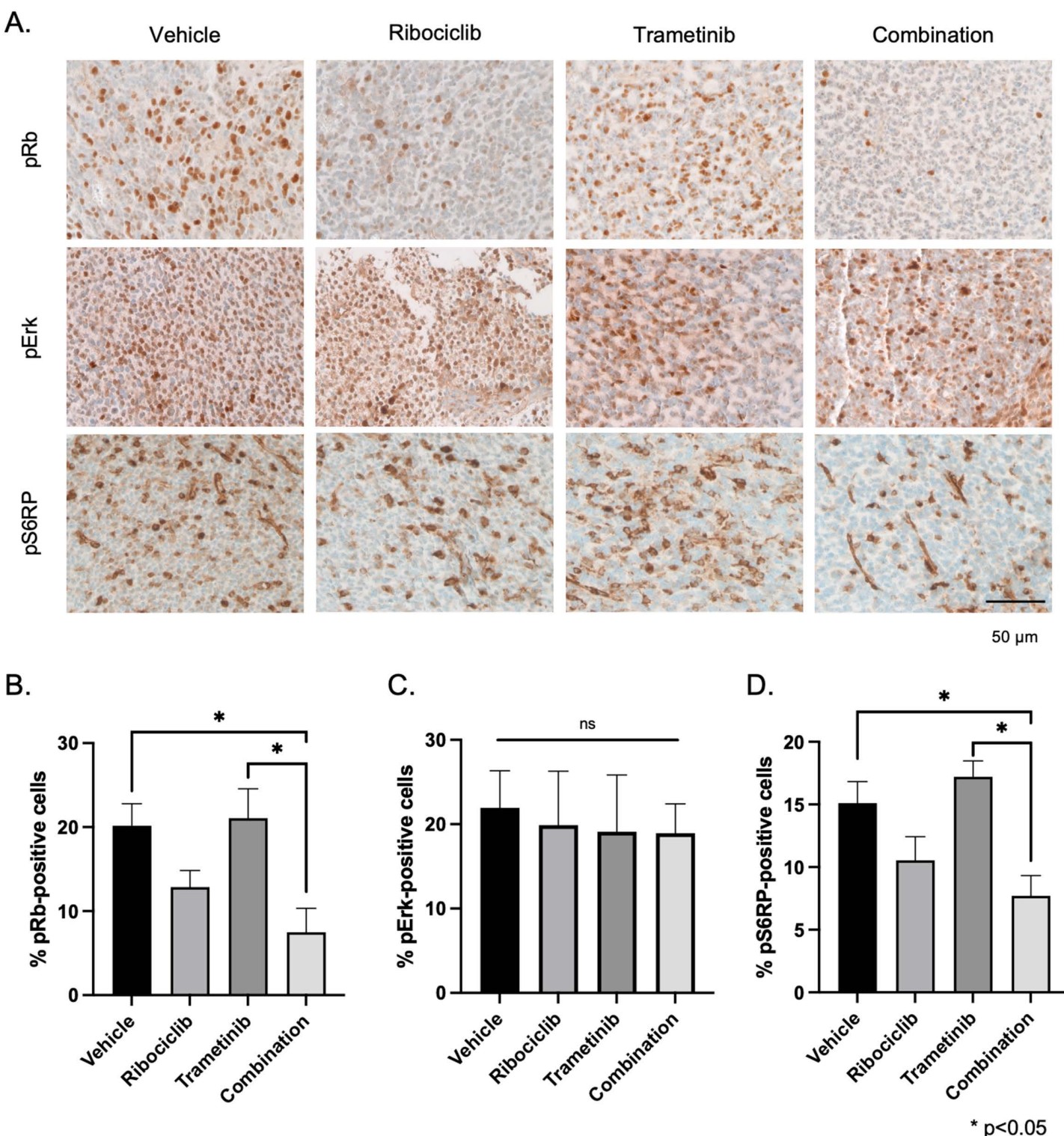

**Fig 6. Effects of short-term treatment on cell proliferation and apoptosis.** A. Representative photos of phospho-Rb, phospho-Erk, phospho-S6 ribosomal protein immunohistochemistry. B. Quantification of phospho-Rb (B), phospho-Erk (C), and phospho-S6 ribosomal protein (D). Combination therapy significantly decreased phospho-Rb- and phospho-S6 ribosomal protein-positive cells compared with vehicle. Pairwise comparison; *p<0.05. Statistical significance was calculated by analysis of variance with 1-way ANOVA with Tukey's post hoc test. Rb, retinoblastoma protein; pS6RP, phospho-S6 ribosomal protein.

p = 0.0306; combination vs. trametinib, p = 0.0201, ANOVA with Tukey's post hoc test) (Fig 6B). Surprisingly, phospho-Erk was not significantly reduced by treatment with trametinib or with combination therapy (Fig 6C), suggesting that our DMG model was intrinsically resistant to trametinib *in vivo*. Previous research has shown that compensatory cross-regulation of Erk and AKT/mTOR signaling may allow glioblastoma cells to proliferate unabated and escape cell death from trametinib treatment [31]. Therefore, we examined the effects of trametinib, ribociclib, and trametinib and ribociclib combined on the AKT/mTOR signaling pathway by quantifying S6 ribosomal protein phosphorylation (pS6RP) changes in response to drug treatments with immunohistochemistry (Fig 6A, 6D). Ribociclib and trametinib as monotherapies did not significantly change the pS6RP expression (ribociclib, p = 0.2289, trametinib; p = 0.8193, ANOVA with Tukey's post hoc test). Only the combination therapy significantly decreased pS6RP expression compared with vehicle (p = 0.0369, ANOVA with Tukey's post hoc test). In summary, combination therapy with ribociclib and trametinib shows additive effects by inhibiting both the RB pathway and the mTOR pathway.

### The impact of combination therapy on the survival of DMG-bearing mice

The anti-tumor effect of the ribociclib and trametinib combination was tested in genetically engineered murine DMGs. From postnatal day 50, we treated mice with ribociclib and trametinib via oral administration for 21 consecutive days (Fig 7A). 16 mice received vehicle, 18 received ribociclib, 16 received trametinib, and 18 received a combination of ribociclib and trametinib. Of all the mice, 21-day treatment was completed in 9 mice treated with vehicle, 12 treated with ribociclib, 9 treated with trametinib, and 15 treated with ribociclib and trametinib combined. Interestingly, the vehicle group's body weight did not change significantly over the 21-day treatment (ADPC: 0.2 [95% CI: −0.5, 0.9]), whereas the body weight of the ribociclib, trametinib, and combination therapy groups significantly increased over the same period (ADPC: 0.6 [95% CI: 0.4, 0.8], 0.4 [95% CI: 0.1, 0.7], and 0.3 [95% CI: 0.0, 0.5], respectively) (Fig 7B). There were no significant differences in body weights among the treatment groups (S1 Table). Brainstem tumors of 66/68 (97.1%) mice were confirmed macroscopically at the time of endpoint, and the remaining two were confirmed microscopically with H&E staining and IHC staining for Ki-67 (S1 Fig). The survival times of mice treated with vehicle, ribociclib, trametinib, or combination therapy were compared by Kaplan–Meier analysis, in which median survivals were 71.5 days, 86.5 days, 72.5 days, and 112 days, respectively (Fig 7C). Combination therapy significantly prolonged mice survival compared with vehicle (p = 0.0195, log-rank test), with ribociclib (p = 0.0477, log-rank test), and with trametinib (p = 0.0057, log-rank test). Three mice treated with combination therapy were alive at six months, although they still had tumors, suggesting that there was a subset group in which combination therapy had a continuous anti-tumor effect.

To evaluate the effect of combination therapy against human DMGs, we conducted an *in vivo* experiment with patient-derived xenograft that was generated by injecting 2x10$^5$ luciferized BT-245 cells into 6-week-old athymic mice. Once tumor engraftment was confirmed via a bioluminescent signal quantified by IVIS, 21-day daily treatment was administered with drug holidays over the weekend. Surprisingly, combination therapy did not significantly prolong mice survival compared with vehicle, ribociclib, or trametinib (S2A Fig). Also, bioluminescence in the mice treated with combination therapy was consistently like those treated with vehicle, ribociclib, or trametinib, even ten days after starting treatment (S2B Fig). These results show that the impact of combination therapy on tumor volume and on mice survival was limited.

### Exploration of the mechanisms of resistance to combination therapy

One plausible explanation for the differential *in vivo* results between the GEM and PDX models is that drug penetration into tumor tissues may vary between the models. Therefore, we evaluated drug concentration using mouse and human DMG tissues treated with ribociclib (75 mg/kg/day), trametinib (0.3 mg/kg/day), and combination (S2 Table). LC/MS analysis of tissue extracts revealed detectable but variable concentrations of ribociclib in the tumor tissues treated with ribociclib and with combination therapy (GEM DMG, ribociclib, 19.58 ± 25.45 µM; combination, 1.21 ± 1.38 µM; human PDX, ribociclib, 5.31 ± 1.89 µM; combination therapy, 2.14 ± 1.35 µM), and these concentrations were not significantly different

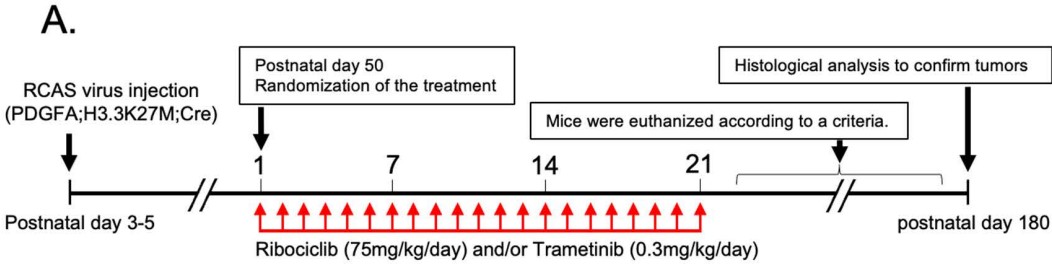

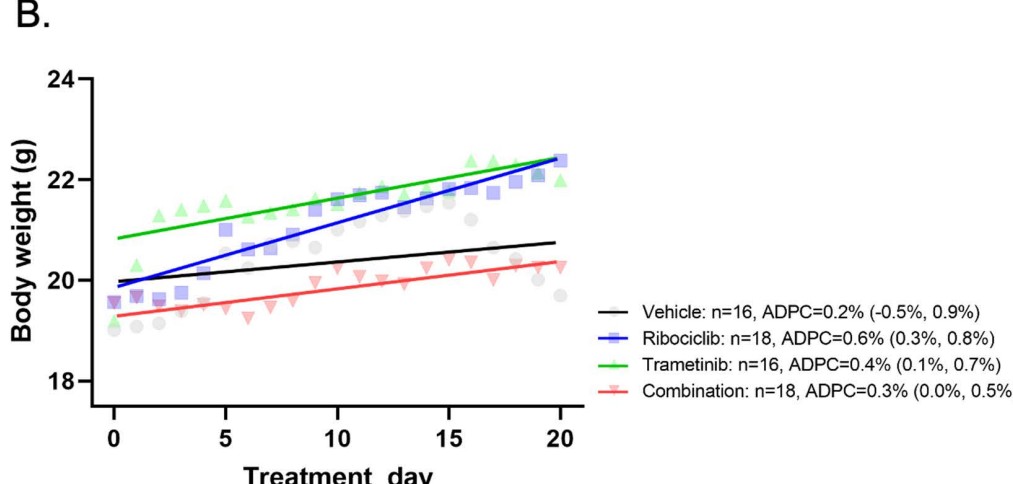

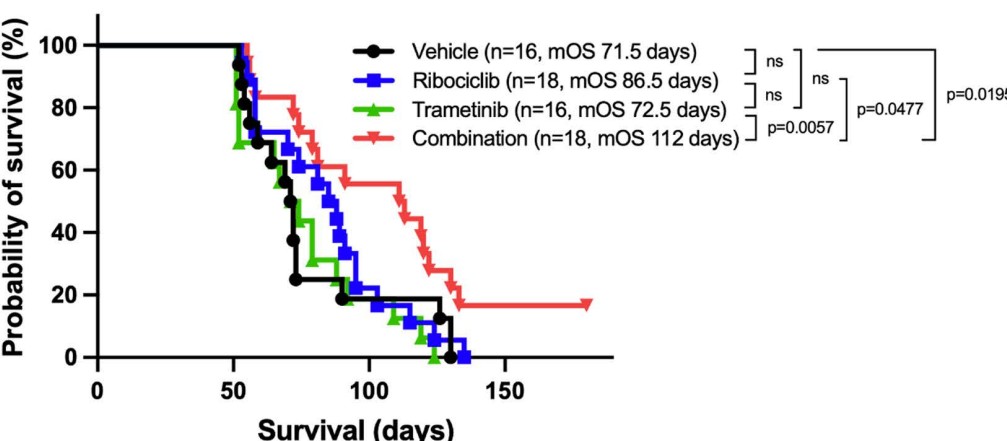

**Fig 7. Evaluation of tumor-bearing mice with long-term treatment.** A. Schematics showing the long-term *in vivo* drug treatment schedule. At post-natal day 50, the animals were randomized into four treatment groups: vehicle, ribociclib alone, trametinib alone, and combination therapy with ribociclib and trametinib. B. Trends in the body weights over 21-day treatment. The body weight of the vehicle group had no apparent trends over the 21-day treatment, whereas that of ribociclib, trametinib, and combination therapy groups significantly increased over the 21 days. C. Kaplan-Meier analysis of mice with long-term treatment. Mice treated with ribociclib and trametinib as combination had a median survival of 112 days, which was substantially longer than that of the vehicle group. Statistical significance was calculated by log-rank test.

(ANOVA test, p = 0.3325) (S3 Fig). Similarly, LC/MS analysis of tissue extracts revealed detectable trametinib in the tumor tissues treated with trametinib and with combination therapy (GEM DMG, trametinib, 0.038 ± 0.014 μM; combination, 0.026 ± 0.020 μM; human PDX, trametinib, 0.037 ± 0.037 μM; combination, 0.040 ± 0.032 μM), and these concentrations were not significantly different (ANOVA test, p = 0.819) (S3 Fig). A possible explanation for the broad range of ribociclib concentrations is the relatively shorter half-life of ribociclib (4.67 hours), while trametinib's half-life increases with repeated doses with a half-life of 33 hours after 7 days of administration [27,32]. Nevertheless, there was no significant difference in ribociclib or trametinib concentrations between monotherapy and combination therapy, indicating that poor drug penetration was not the primary mechanism for the limited efficacy in the human PDX model.

Next, to evaluate the effects of ribociclib and trametinib on the transcriptome in the survival cohort, RNA-seq analysis was performed with DMG tissue harvested from mice in the survival study (GEM model) treated with ribociclib, trametinib, combination therapy, or vehicle (S3 Table). When we compared the transcriptome of ribociclib-treated mice to vehicle-treated mice, we found only 13 significantly differentially expressed genes, of which 5 were upregulated and 8 were downregulated (S4 Table). By contrast, comparison of the transcriptome of trametinib-treated mice to vehicle-treated mice identified 124 significantly differentially expressed genes, including 48 upregulated and 76 downregulated genes (S5 Table). Interestingly, comparison of the transcriptome of combination-treated mice to vehicle-treated mice identified 905 significantly differentially expressed genes, including 381 upregulated and 524 downregulated genes (Fig 8A, S6 Table). Principal component analysis and unsupervised hierarchical clustering showed that tumors treated with combination therapy clustered separately from those treated with vehicle (Fig 8B and 8C).

Next, we performed gene set enrichment analysis (GSEA) to evaluate if the inhibitors downregulated the expected pathways (S7-12 Table). The KRAS signaling pathway was significantly downregulated by all the treatment groups compared with the vehicle group. As an example, *Ccnd2 (Cyclin D2)*, a member of the Hallmark Kras upregulated gene-set, was one of the genes significantly downregulated by ribociclib while *Gabra3*, also a member of the Hallmark Kras upregulated gene-set, was one of the genes significantly downregulated by trametinib. Several Hallmark Kras upregulated genes were identified among the large number of significantly differentially expressed genes in the comparison between combination therapy and vehicle groups: *Ccnd2, Mmp10, Mmp9, Ets1, Hdac9, Cbl, and Pecam1*, highlighting that the combination had strong transcriptomic effect on the Kras pathway.

We also observed that the GSEA analysis was consistent with the immunohistochemistry analysis of the short-term treatment cohorts in the GEM model, with trametinib having a limited effect on proliferation and the mTOR pathway. Hallmark_E2F_Targets (NES = 2.7, p < 0.0001), Hallmark_Myc Targets_V2 (NES = 2.33; p < 0.0001), Hallmark G2M_Checkpoint (NES = 2.2, p < 0.0001), and Hallmark_MTORC1 (NES = 1.5, p < 0.03) signaling were significantly positively enriched by trametinib treatment relative to vehicle treatment. By contrast, Hallmark_MTORC1 (NES = −1.79; p = 0.001) was inhibited by the combination therapy. This aligns with the limited effects of trametinib on cell proliferation or the mTOR pathway. Interestingly, we identified a significant correlation between combination therapy and downregulation of genes in the "KEGG_GLIOMA" gene set, which includes key drivers of gliomagenesis such as CDK4/6, PDGFA/PDGFB/PDGFRA/PDGFRB/EGF/EGFR (Fig 8D) and of genes in the "PI3K_AKT_MTOR_Signaling" gene set (Fig 8E). We also conducted IHC for Ki-67 and cleaved caspase-3 to evaluate the cytostatic and cytotoxic effects of long-term treatment with ribociclib and trametinib in combination (S4A-S4B Fig). Combination therapy did not change the Ki-67-positive rate compared with that of the vehicle (combination, 11.1 ± 2.6%, vehicle, 9.1 ± 2.5%, p = 0.5892, Student's *t* test). In contrast, combination therapy significantly increased cleaved caspase-3-positivity relative to that of the vehicle (combination, 2.5 ± 0.7%, vehicle, 0.8 ± 0.2%, p = 0.0376, Student's *t* test). These results suggest that cytotoxic effects, but not cytostatic effects, were sustained after 21-day treatment with ribociclib and trametinib combined. GSEA analysis also found that tumors treated with combination therapy showed significant negative enrichment in the gene sets of inflammatory responses compared with tumors that received vehicle (Fig 8F). Interestingly, we observed significant changes in genes that are primarily expressed in myeloid cells: for example, *Myd88* was upregulated by the combination, and Il18 bp were downregulated. To investigate

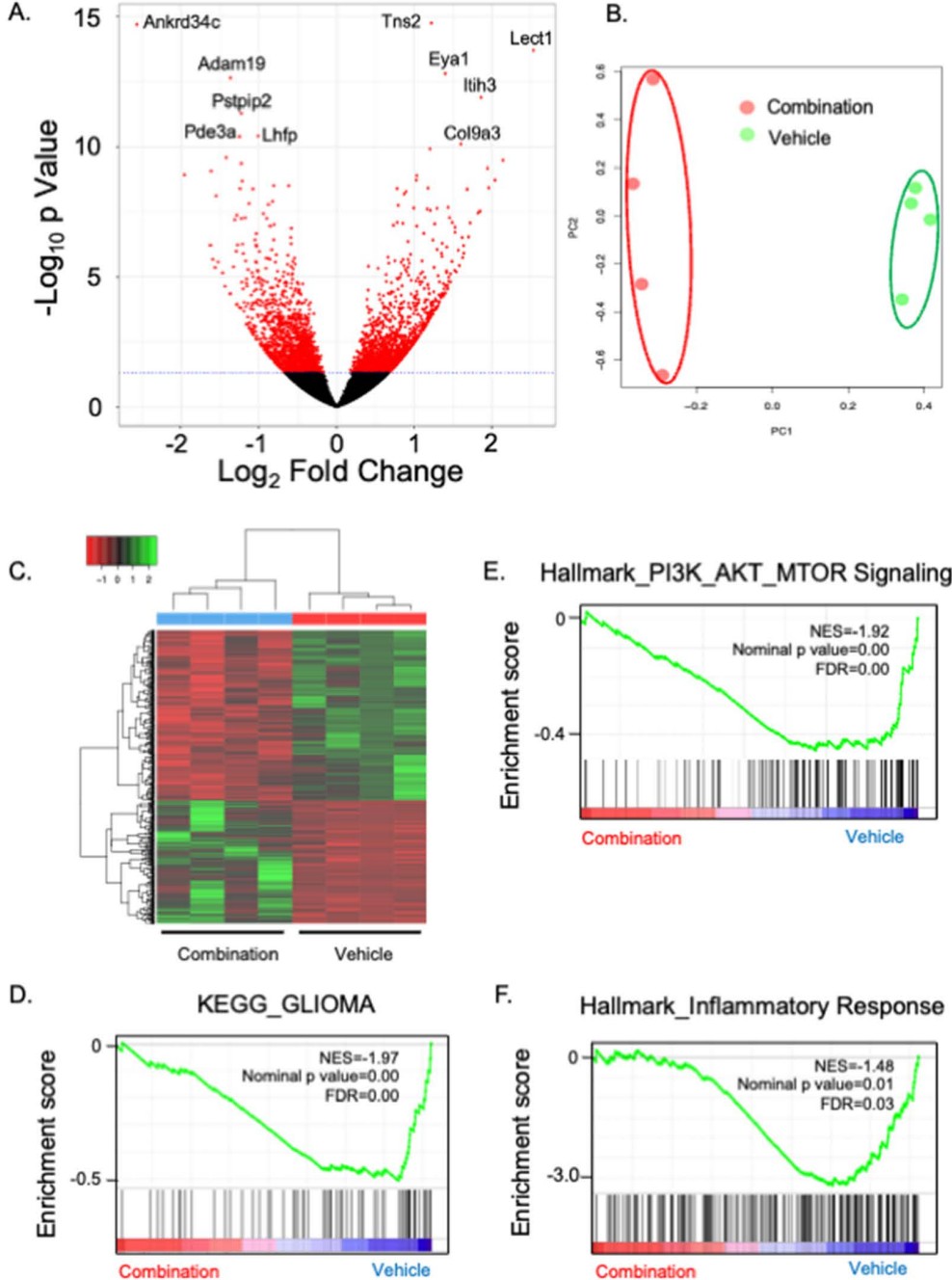

**Fig 8. Transcriptomic analysis of DMG tumors treated with combination therapy or with vehicle.** A. Frozen tissues for RNA-sequencing were extracted at the time mice reached their endpoint. Volcano plot comparing gene expression of mice tumors treated with combination therapy and those that received vehicle. There was a total of 905 significantly differentially expressed genes, including 381 upregulated and 524 downregulated, in the combination group compared with the vehicle group. n = 4. B. Principal component analysis showing that the tumors treated with combination therapy and those that received vehicle were clearly segregated. C. Unsupervised hierarchical clustering of differentially regulated genes (Padj<0.05) between tumors treated with vehicle and those treated with combination (n = 4 each). D. GSEA plot for "KEGG_GLIOMA" gene set in the combination therapy group relative to the vehicle group. E. GSEA plot for PI3K/AKT/mTOR signaling gene set in the combination therapy group relative to the vehicle group. F. GSEA plot for inflammatory response gene set in the combination therapy group relative to the vehicle group.

if the potential immune effects impacted the number of lymphocytes, we performed immunohistochemistry staining for CD3, CD4, and CD8 (S5A Fig). Although tumor tissues treated with ribociclib or trametinib as monotherapies tended to show more CD3- and CD4-positive cells, combination therapy did not change CD3/CD4/CD8 expressions compared with those of tumors that received vehicle (S5B-S5D Fig). This suggests that the transcriptomic effects on inflammation may be due to myeloid cells rather than lymphocytes.

## Discussion

Ribociclib is one of several FDA-approved CDK4/6 inhibitors. A recent phase I/II study suggests that this drug is well tolerated in children with DMG [11]. However, CDK4/6 inhibitors should be used as combination chemotherapy rather than as monotherapy [13]. Trametinib is a well-known MEK1/2 inhibitor commonly used in combination with dabrafenib to treat melanoma and BrafV600E mutant pediatric gliomas. There is an ongoing clinical trial at St. Jude's to evaluate the combination therapy with ribociclib and trametinib for recurrent central nervous system tumors including "diffuse midline glioma, *H3K27M*-mutant" (NCT03434262) that suggests that the combination can be well tolerated [33]. However, there is no report to investigate the combination therapy of a CDK4/6 inhibitor with a MEK inhibitor in preclinical DMG models.

By contrast, there have been multiple preclinical studies evaluating the efficacy of a CDK4/6 inhibitor in combination with a MEK inhibitor in other types of cancer. First, there was a preclinical study in melanoma demonstrating that the combination of CDK4/6 plus MEK inhibitors has greater efficacy in immunocompetent models relative to immunodeficient models, and that CD8＋T cells contribute to the antitumor effects of this combination [34]. In a second study, the combination of CDK4/6 and MEK inhibition resulted in activation of interferon pathways while in a third study the combination resulted in upregulation of oxidative phosphorylation genes [35,36]. More recently, a preclinical study in malignant peripheral nerves sheath tumors or MPNSTs noted that treatment with the combination leads to plasma cell infiltration, sensitization to PD-L1 blockade and tumor regression in an immunocompetent model [37]. With regards to clinical evaluation of the combination of CDK4/6 and MEK inhibition in patients with cancer, there was a phase Ib in unselected adult patients with solid tumors that did not observe any efficacy [38], while a case series of nine adult patients with solid tumor harboring CDK4/6 and MEK pathway alterations treated with CDK4/6 and MEK inhibitors noted some clinical benefit in approximately 50% of patients [39]. Therefore, the rationale for this preclinical study was to investigate the efficacy and the mechanism of this combination therapy in an immunocompetent GEM DMG model and an immunodeficient PDX DMG model.

We used human DMG cell lines for *in vitro* synergy studies, and both GEM DMG and PDX DMG models for *in vivo* experiments. Conducting *in vitro* experiments with human cell lines is a useful first step to examine if the drug treatments are promising. By contrast, for *in vivo* experiments, genetically engineered models may be more relevant preclinical models for DMG compared with human-derived xenograft models, because the former can recapitulate the genetic, biochemical, and phenotypic features of DMG in an immunocompetent host, with new evidence that gliomagenesis itself induces systemic immunosuppression [40,41]. Our autochthonous model evolves *de novo* in each mouse, with three genetic drivers developing a unique tumor due to variable integration of the drivers at different copy numbers and in different genomic loci [42]. We have previously reported the preclinical study of palbociclib using a murine DMG model with platelet-derived growth factor-B (PDGFB) overexpression and *INK4A-ARF* deletion [8]. With a tumor formation rate as high as 90%, our model was feasible as a preclinical trial to evaluate the efficacy of this novel combination therapy.

Our *in vivo* data suggests that the effects of ribociclib are mainly cytostatic rather than cytotoxic. By contrast, trametinib increased the number of cleaved caspase-3-positive cells, so it had a cytotoxic effect in a subset of cells, but it did not significantly impact proliferation based on the immunohistochemistry and RNAseq results. Most importantly, trametinib reduced pErk levels *in vitro* but not *in vivo*. These results imply that our genetically engineered DMG model has intrinsic resistance to trametinib despite the adequate delivery of trametinib to the tumor tissues. Our observations are consistent with observations by others in other cancer models [43]. Kwong et al. noted that trametinib monotherapy induced

apoptosis, but not cell-cycle arrest, in a mutant *Nras* model of melanoma. Additionally, Schreck et al. investigated the effects of trametinib on glioblastoma cell lines *in vitro*, showing that ERK phosphorylation was inhibited by trametinib one hour after treatment but began to recover by 24 hours [31]. Thus, our DMG model could acquire resistance to trametinib within a five-day treatment, as we dosed trametinib only once a day. Schreck et al. has also shown that pS6RP might predict sensitivity to trametinib [31]. Phospho-S6 ribosomal protein upregulates cell cycle checkpoint proteins such as pRb and CDK6 [44], well-known targets of CDK4/6 inhibitors. Moreover, CDK4/6 inhibitors could downregulate pS6RP in *KRAS*-mutant colorectal cancers [45]. Consistent with those observations, our data suggests that pS6RP may be associated with resistance to trametinib in our model, and the additive effects of combination therapy may have been due to downregulation of pS6RP by adding ribociclib. To our knowledge, this is the first report showing the effects of combination therapy with a MEK inhibitor and a CDK4/6 inhibitor in DMG. Further investigation is required to clarify the mechanism of the additive effects observed by the combination of ribociclib and trametinib in our DMG model. In addition, it would be interesting to test higher doses of trametinib in combination with ribociclib.

In line with the synergistic effects shown by *in vitro* assays and with the short-term effects of the combination therapy in DMG-bearing mice, longer treatment for 21 days showed a significant effect on survival. This suggests that the combination therapy has at least additive effects *in vivo* in an immunocompetent model. Interestingly, our RNAseq data showed that several signaling pathways, including the cell proliferation pathway, the mTOR pathway, and the DNA repair pathway, were not upregulated even after cessation of long-term combination therapy, compared with the vehicle. Also, immunohistochemical analysis of the tumor tissue after cessation of the long-term treatment showed that combination therapy still maintained cytotoxic effects, but not cytostatic effects, compared with the vehicle. These results imply that short-term treatments with trametinib may be enough to have long-term cytotoxic effects, but treatment with ribociclib needs to be maintained indefinitely. As for the immunogenic changes, our GSEA analysis showed that combination therapy reduced the expression of gene sets related to inflammatory responses, significantly impacting genes expressed in myeloid cells. As previous reports showed that a combination of MEK and CDK4/6 inhibitors stimulated the accumulation of CD-8 + T-cells [46], immunological changes from combination therapy may differ between tumor types. A recently completed phase 1 clinical trial to evaluate combination therapy with ribociclib and trametinib (NCT03434262) in 28 patients with refractory/relapsed CNS tumors (13 of which were high-grade glioma) suggests that this combination is tolerable in patients and may have anti-tumor activity in select patients [33].

In this preclinical research, we evaluated the combination therapy for only 5–21 days, noting limited activity of both ribociclib and trametinib as single agents, and moderate *in vivo* activity in combination in the aggressive murine GEM DMG model. Although therapeutic effects seem modest, longer-term treatment may have a more pronounced effect against DMGs, or a third drug may be needed to overcome resistance to the two drugs. For example, both Id3 and Id4 were significantly upregulated in the combination cohort relative to vehicle, so these proteins may confer resistance to therapy and should be studied in the future. Also, short-term treatment showed that the combination therapy did not significantly decrease cell proliferation compared with ribociclib or trametinib as monotherapies. There are several plausible explanations for these results: first, statistical underpower due to the lower number of mice used for IHC; second, the inherent heterogeneity of the model, as each tumor is induced *de novo*; and third, the different mechanisms of action of the two drugs, which have different effects on cell proliferation and survival.

There are several limitations to our study. First, as DMG patients are usually diagnosed by magnetic resonance imaging (MRI), performing MRIs of tumor-bearing mice before and after treatment initiation could improve the rigor of our study and allow us to assess if the combination therapy was potent enough to reduce the size of any of the tumors. Second, we did not conduct a thorough assessment of systemic toxicity in tumor-naïve mice. Therefore, it would be interesting to learn about the toxicity profile from the recently completed clinical trial in children with DMG. The dosing was chosen based on a recommendation from Novartis for human-equivalent dosing in mice, suggesting that these doses can be safely tolerated in humans at least as single agents. Future studies should also evaluate the safety profile of this combination therapy

in tumor-bearing mice. Third, our data demonstrates that combination therapy with ribociclib and trametinib is effective against our GEM DMG model driven by PDGF signaling, H3.3K27M and p53 loss, and showing histologically high-grade H3.3K27M-positive tumors, but not the DMG PDX which harbors PDGFRA mutations, H424Y and P443L, TP53 R249S mutation, CDKN2A/B deletion, and MYC amplification [47]. Future studies will elucidate the mechanisms for this discrepancy. One potential explanation is that MYC amplification may confer resistance to this combination as the study by Hart et al. noted that MYCN amplification is associated with resistance to MEK inhibition [28]. Another potential explanation for the lack of efficacy in the PDX model is the testing in an immunodeficient model as multiple groups have noted greater efficacy of the combination in immunocompetent models and/or an immune mechanism as part of the anti-tumor effect of this combination [34,35,37].

## Conclusions

Our results across both human and mouse DMG models demonstrate that combination therapy with ribociclib and trametinib has additive cytostatic and cytotoxic effects, and a significant effect on the survival of DMG-bearing mice in the GEM model but not in the PDX model. The differential results between the GEM and PDX models highlight the importance of testing novel therapies in diverse models and comparing results to the gold standard- anti-tumor activity in patients. Using a *de novo* model in which each mouse develops a unique tumor driven by the same three genetic drivers increases rigor, factoring in tumor heterogeneity, which is one of the hallmarks of cancer in general.

We conclude that this combination should be tested upfront in a clinical trial in DMG tumors harboring PDGFRA and H3.3K27M alterations.

### Key Points

1. Combination therapy had cytotoxic and cytostatic effects *in vivo* in the GEM model.

2. Combination therapy had variable synergy *in vitro* in human DMG models.

3. Combination therapy significantly prolonged mice survival compared with the vehicle in the GEM model.

4. Combination therapy did not prolong survival in the PDX model.

5. Transcriptional analysis suggests that the combination modulated inflammation in the GEM model.

6. This work highlights the importance of testing novel therapies in diverse models.

## Supporting information

**S1 Fig. Macroscopic and microscopic images of the mice that survived at postnatal day 180 after combination therapy.**
(TIFF)

**S2 Fig. In vivo experiment with patient-derived xenograft.**
(TIFF)

**S3 Fig. LC/MS analysis of tissue extracts from tumor tissue after treatment with ribociclib and/or trametinib.**
(TIFF)

**S4 Fig. Immunohistochemistry staining for Ki-67 and cleaved caspase-3 of tumors after 21-day treatment.**
(TIFF)

**S5 Fig. Immunohistochemistry staining for CD3, CD4, and CD8 of tumors after 21-day treatment.**
(TIFF)

**S6 Fig. Raw data -uncropped western blots from Fig 1D.**
(PDF)

**S1 Table. Background information regarding the DMG GEM mice used in 21-day survival study.**
(DOCX)

**S2 Table. LC/MS analysis of ribociclib and trametinib in DMG tumors.**
(DOCX)

**S3 Table. Background information of the DMG GEM mice used for RNAseq.**
(DOCX)

**S4 Table. Differentially expressed genes between tumors treated with ribociclib and those treated with vehicle.**
(DOCX)

**S5 Table. Differentially expressed genes between tumors treated with trametinib and those treated with vehicle.**
(DOCX)

**S6 Table. Differentially expressed genes between tumors treated with combination and those treated with vehicle.**
(DOCX)

**S7 Table. GSEA analysis comparing tumors treated with ribociclib and those treated with vehicle.**
(DOCX)

**S8 Table. GSEA analysis comparing tumors treated with trametinib and those treated with vehicle.**
(DOCX)

**S9 Table. GSEA analysis comparing tumors treated with combination and those treated with vehicle.**
(DOCX)

**S10 Table. GSEA analysis comparing tumors treated with ribociclib and those treated with trametinib.**
(DOCX)

**S11 Table. GSEA analysis comparing tumors treated with combination and those treated with ribociclib.**
(DOCX)

**S12 Table. GSEA analysis comparing tumors treated with combination and those treated with trametinib.**
(DOCX)

**S13 Table. Cell-lines used in the study.**
(PDF)

## Acknowledgments

We acknowledge the contribution of Dr. Yoshihiro Tanaka who provided statistical expertise in how to analyze the data in this study, particularly, the analysis of the trends in body weights in the GEM preclinical survival study

## Author contributions

**Conceptualization:** Oren J Becher.

**Data curation:** Yusuke Tomita, Yi Ge, Jun Watanabe, Rintaro Hashizume.

**Formal analysis:** Yusuke Tomita, Samantha Gadd.

**Funding acquisition:** Oren J Becher.

**Investigation:** Yusuke Tomita, Gabrielle Link, Yi Ge, Herminio Joey Cardona, Megan Romero, Samantha Gadd, Jun Watanabe, Eita Uchida, Rintaro Hashizume, Nozomu Takata, Gonzalo Pinero, Dolores Hambardzumyan, Ivan Spasojevic, Guo Hu, Tammy Hennika, Daniel J Brat, Adam L Green.

**Methodology:** Anna Racanelli, Oren J Becher.

**Project administration:** Oren J Becher.

**Resources:** Herminio Joey Cardona, Megan Romero.

**Supervision:** Oren J Becher.

**Visualization:** Samantha Gadd.

**Writing – original draft:** Yusuke Tomita.

**Writing – review & editing:** Emma R.H. Gold, Oren J Becher.

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
