## [Decision Letter · Decision Letter 0]

27 Jun 2025

Dear Dr. Becher,

Thank you for submitting your manuscript to PLOS ONE. After careful consideration, we feel that it has merit but does not fully meet PLOS ONE’s publication criteria as it currently stands. Therefore, we invite you to submit a revised version of the manuscript that addresses the points raised during the review process.

**Comments from the academic editor are found below.**

We look forward to receiving your revised manuscript.

Kind regards,

Marielle Yohe

Academic Editor

PLOS ONE

Journal Requirements:

“This work is supported by the B+ McDonough Foundation, Cristian Rivera Foundation, Madox's Warriors, and DIPG Collaborative to OJB. OJB and DH are supported by R01CA258636”

4. Please include captions for your Supporting Information files at the end of your manuscript, and update any in-text citations to match accordingly. Please see our Supporting Information guidelines for more information: http://journals.plos.org/plosone/s/supporting-information .

Additional Editor Comments:

In this manuscript, Dr. Becher and colleagues report on a beautiful preclinical study evaluating the efficacy of the combination of a CDK4/6 inhibitor (ribociclib) and a MEK inhibitor (trametinib) in models of diffuse midline glioma (DMG). DMG is a deadly form of pediatric brain cancer for which there are no FDA approved therapies. Radiation therapy is provided upfront but all patients eventually relapse and die of their disease, usually within a year from diagnosis. Therefore, DMG is an area of acute unmet medical need and the current study provides a potential treatment avenue for these patients. Overall, the study is well conducted and well controlled. I have several comments/questions about the interpretation of the data and how to better place the data into the context of the literature.

1) The authors' beautifully describe the biomarker that allows for DMG sensitivity to CDK4/6 inhibition, the loss of p16. However, they don't discuss potential biomarkers providing for sensitivity to MEK inhibition (ie RAS/MAPK pathway SNVs or fusions). Could the authors describe this, and as well provide a table of the molecular alterations in the cell lines used in the study?

2) In prior studies, trametinib has not crossed the blood brain barrier. It is interesting that the authors see significant trametinib exposure in both the tumor and the cerebral cortex in the DMG model. Does this indicate that the BBB is not intact in tumor bearing mice? Additionally the authors mention that the half life of trametinib is 4 days, which is true in humans but it is much shorter (24 h) in mice. I think too it would be good to point out that in figure 1 the dose of trametinib being used is 10X the amount used in the efficacy study, and the exposure in the efficacy study is 1/10th that seen in figure 1 - which is right about the IC50 of trametinib, not several fold higher as is discussed earlier in the paper. This could account for some of the resistance seen in the PDX model. Could the authors repeat the western blot in Figure 1D/E with a lower concentration of trametinib? 10 uM is quite high and not clinically achievable.

3) As reviewer 1 points out, the MEKi/CDK4/6i combo has been used in several other studies. Particularly relevant to this paper would be another pediatric cancer, neuroblastoma (Hart LS, Rader J, Raman P, Batra V, Russell MR, Tsang M, Gagliardi M, Chen L, Martinez D, Li Y, Wood A, Kim S, Parasuraman S, Delach S, Cole KA, Krupa S, Boehm M, Peters M, Caponigro G, Maris JM. Preclinical Therapeutic Synergy of MEK1/2 and CDK4/6 Inhibition in Neuroblastoma. Clin Cancer Res. 2017 Apr 1;23(7):1785-1796. doi: 10.1158/1078-0432.CCR-16-1131. Epub 2016 Oct 11. PMID: 27729458.) If the authors could broaden their literature review of the combination that would be great.

Reviewers' comments:

Reviewer's Responses to Questions

**Comments to the Author**

1. Is the manuscript technically sound, and do the data support the conclusions?

Reviewer #1: Yes

2. Has the statistical analysis been performed appropriately and rigorously?

Reviewer #1: Yes

3. Have the authors made all data underlying the findings in their manuscript fully available?

Reviewer #1: Yes

4. Is the manuscript presented in an intelligible fashion and written in standard English?

Reviewer #1: Yes

Reviewer #1: Very well conceptualised and written study.

1. Was toxicity of this combination assessed or will there be further studies planned to assess it?

2. Ribociclib and trametinib combination has already been assessed in clinical trials and been found to be not effective (LoRusso et al Annals of Oncology 2020). Based on these results, do the authors think this combination can be investigated in the setting of a clinical trial? Authors have not mentioned this in their discussion or conclusion.

4. Considering all the results, do the authors feel ribociclib is adding any value to tramertinib's activity?

**Do you want your identity to be public for this peer review?** For information about this choice, including consent withdrawal, please see our Privacy Policy

Reviewer #1: **Yes: ** Dr Santosh Valvi

---

## [Author Response · Author response to Decision Letter 1]

5 Sep 2025

Response letter

To the editor:

We would like to thank the editors and reviewers for their thorough review. We have made edits and additional experimentation in response to the reviewer comments and as a result the manuscript is significantly improved. Please note that all the experiments for the revision were done by Anna Racanelli in the lab and so we are adding her as a co-author.

We responded to each comment from Reviewers in a point-by-point manner.

Editor: In this manuscript, Dr. Becher and colleagues report on a beautiful preclinical study evaluating the efficacy of the combination of a CDK4/6 inhibitor (ribociclib) and a MEK inhibitor (trametinib) in models of diffuse midline glioma (DMG). DMG is a deadly form of pediatric brain cancer for which there are no FDA approved therapies. Radiation therapy is provided upfront but all patients eventually relapse and die of their disease, usually within a year from diagnosis. Therefore, DMG is an area of acute unmet medical need and the current study provides a potential treatment avenue for these patients. Overall, the study is well conducted and well controlled. I have several comments/questions about the interpretation of the data and how to better place the data into the context of the literature.

1) The authors' beautifully describe the biomarker that allows for DMG sensitivity to CDK4/6 inhibition, the loss of p16. However, they don't discuss potential biomarkers providing for sensitivity to MEK inhibition (ie RAS/MAPK pathway SNVs or fusions). Could the authors describe this, and as well provide a table of the molecular alterations in the cell lines used in the study?

This is a great point raised by the editor. The models used for the in vivo studies are mainly driven by PDGF signaling which activates the RAS/MAPK pathway. There is no specific mutation or fusion in RAS or RAF or MEK or ERK, just a high level of PDGFRA. The human line we used BT-245 has PDGFRA mutations, p53 mutations, MYC amplification, and CDKN2A/B loss[1]. Based on the Hart paper[2], it may be that the resistance of BT-245 to the combination of MEK/CDK4-6 inhibitors may be at least in part mediated by the MYC amplification. Below is a table of the cell-lines used in the study which we be Supplemental Table 13.

14-1214-1, 14-1206-1, 14-1206-5 (murine) PDGFB; H3.3K27M; p53 loss

23-0509-2 (murine) PDGFB; H3.3K27M; p53 loss

23-0104-3 (murine) PDGFB; H3.3K27M; p53 loss

4738 (murine) PDGFB; H3.3K27M; p53 loss

BT-245 (human) H3.3K27M; H424Y and P443L mutations in PDGFRA; TP53.R249S; MYC amplification, CDKN2A/B loss

SF8628 (human) H3.3K27M

SF7761 (human) H3.3K27M

HSJD-DIPG-007 (human) H3.3K27M; ACVR1 R206H; PPM1D Pro428Ginfs*3

SU-DIPG17 (human) H3.3K27M

2) In prior studies, trametinib has not crossed the blood brain barrier. It is interesting that the authors see significant trametinib exposure in both the tumor and the cerebral cortex in the DMG model. Does this indicate that the BBB is not intact in tumor bearing mice? Additionally the authors mention that the half life of trametinib is 4 days, which is true in humans but it is much shorter (24 h) in mice. I think too it would be good to point out that in figure 1 the dose of trametinib being used is 10X the amount used in the efficacy study, and the exposure in the efficacy study is 1/10th that seen in figure 1 - which is right about the IC50 of trametinib, not several fold higher as is discussed earlier in the paper. This could account for some of the resistance seen in the PDX model. Could the authors repeat the western blot in Figure 1D/E with a lower concentration of trametinib? 10 uM is quite high and not clinically achievable.

Thank you for your insightful comments and suggestions. Yes, the most plausible explanation for LC/MS results is that the BBB is not intact in the tumor-bearing models. Of note, trametinib + dabrafenib are FDA approved to treat BRAF V600E mutant gliomas and so it has sufficient BBB penetration to be active against some types of brain tumors[3]. We repeated the westerns in Figure 1D/E with 1uM, and the results are pasted below (see official response to reviewer comments word document for the western and quantification of the data). The results were similar to the higher dose of 10uM which we agree was too high and not clinically achievable, so we appreciate the reviewer asking us to repeat the western at a lower dose.

The trametinib drug levels in the GEM and PDX tumors were similar with a range of 0.026-0.038uM for the GEMs, and 0.037-0.040uM for the PDX and in a similar range to the IC50s of the murine and human DMG cell-lines: range of IC50s in the murine DMG lines was 0.033-0.058uM, range of IC50s in the human DMG lines was 0.004-0.117uM. Please note that the lower doses of trametinib (0.3mg/kg/day) were used for the survival study and the drug measurements of those mice had trametinib levels that were similar to the IC50s in Figure 1. We also corrected the information regarding the half-life of trametinib and ribociclib in the text on lines 560-562, page 25-26. Trametinib has a half-life of 33 hours after 7 days of administration in mice[4] while ribociclib has a half-life of 4.67 hours[5].

3) As reviewer 1 points out, the MEKi/CDK4/6i combo has been used in several other studies. Particularly relevant to this paper would be another pediatric cancer, neuroblastoma (Hart LS, Rader J, Raman P, Batra V, Russell MR, Tsang M, Gagliardi M, Chen L, Martinez D, Li Y, Wood A, Kim S, Parasuraman S, Delach S, Cole KA, Krupa S, Boehm M, Peters M, Caponigro G, Maris JM. Preclinical Therapeutic Synergy of MEK1/2 and CDK4/6 Inhibition in Neuroblastoma. Clin Cancer Res. 2017 Apr 1;23(7):1785-1796. doi: 10.1158/1078-0432.CCR-16-1131. Epub 2016 Oct 11. PMID: 27729458.) If the authors could broaden their literature review of the combination that would be great.

Response: Thank you for this important feedback. Indeed, the combination of MEKi/CDK4/6i has been tested in several other studies, and two independent studies have noted that this combination has more antitumor activity in immunocompetent models- reference 1 and 5 below. We added the following references to the manuscript:

1. Activation of CD8+ T Cells Contributes to Antitumor Effects of CDK4/6 Inhibitors plus MEK Inhibitors. doi: 10.1158/2326-6066.CIR-19-0743. Interestingly this paper also noted increased efficacy of this combination in immunocompetent models vs. immunodeficient models in the context of melanoma[6].

2. Separable cell cycle arrest and immune response elicited through pharmacological CDK4/6 and MEK inhibition in RASmut disease models. doi:10.1158/1535-7163.MCT-24-0369. This paper reported activation of interferon pathways in response to inhibition of CDK4/6 and MEK[7].

3. Metabolic Adaptations to MEK and CDK4/6 Cotargeting in Uveal Melanoma. doi: 10.1158/1535-7163.MCT-19-1016. This paper noted upregulation of oxidative phosphorylation in response to MEK and CDK4/6 inhibition in uveal melanoma[8].

4. Concomitant MEK and Cyclin Gene Alterations: Implications for Response to Targeted Therapeutics. doi: 10.1158/1078-0432.CCR-20-3761. This paper reports the results of treating a small number of patient with concomitant CDK4/6 and MAPK pathway alterations with a combination of a MEK and CDK4/6 inhibitor and noting clinical benefit in approximately 50% of the patients[9].

5. CDK4/6-MEK Inhibition in MPNSTs Causes Plasma Cell Infiltration, Sensitization to PD-L1 Blockade, and Tumor Regression. https://doi.org/10.1158/1078-0432.CCR-23-0749. This paper also highlights an immune related mechanism to the response to CDK4/6-MEK inhibitors in MPNSTs. The combination has significantly better antitumor effects than the single agents[10].

Page 29, Line 628-643: “By contrast, there have been multiple studies evaluating the efficacy of a CDK4/6 inhibitor in combination with a MEK inhibitor in other types of cancer. First, there was a preclinical study in melanoma demonstrating that the combination of CDK4/6 plus MEK inhibitors has greater efficacy in immunocompetent models, and that CD8 + T cells contribute to the antitumor effects of this combination [6]. In a second study, the combination of CDK4/6 and MEK inhibition resulted in activation of interferon pathways while in a third study the combination resulted in upregulation of oxidative phosphorylation genes [7, 8]. More recently, a preclinical study in malignant peripheral nerves sheath tumors or MPNSTs noted that the combination leads to plasma cell infiltration, sensitization to PD-L1 blockade and tumor regression in an immunocompetent model[10]. With regards to clinical evaluation of the combination of CDK4/6 and MEK inhibition in patients with cancer, there was a phase Ib in unselected adult patients with solid tumors that did not observe any efficacy[11] while a case series of nine adult patients with solid tumor harboring CDK4/6 and MEK pathway alterations treated with CDK4/6 and MEK inhibitors noted some clinical benefit in approximately 50% of patients [9]. Therefore, the rationale for this study was to investigate the efficacy and the mechanism of this combination therapy in an immunocompetent GEM DMG model and an immunodeficient PDX DMG model.”

Reviewers' comments:

Reviewer's Responses to Questions

Comments to the Author

1. Is the manuscript technically sound, and do the data support the conclusions?

Reviewer #1: Yes

2. Has the statistical analysis been performed appropriately and rigorously?

Reviewer #1: Yes

3. Have the authors made all data underlying the findings in their manuscript fully available?

Reviewer #1: Yes

4. Is the manuscript presented in an intelligible fashion and written in standard English?

Reviewer #1: Yes

5. Review Comments to the Author

Reviewer #1: Very well conceptualized and written study.

1) Was toxicity of this combination assessed or will there be further studies planned to assess it?

Response: Thank you for bringing up the important issue of toxicity. We did not evaluate the toxicity of this combination in this study. Yes, there will be further studies to assess it, particularly if this combination will move forward into a clinical trial with children with DMG.

Page 32, Line 711-716: “Second, we did not conduct a thorough assessment of systemic toxicity in tumor-naïve mice. Therefore, it would be interesting to learn about the toxicity profile from the recently completed clinical trial in children with DMG. The dosing was chosen based on a recommendation from Novartis for human-equivalent dosing in mice, suggesting that these doses can be safely tolerated in humans at least as single agents. Future studies should also evaluate the safety profile of this combination therapy in tumor-bearing mice”

2) Ribociclib and trametinib combination has already been assessed in clinical trials and been found to be not effective (LoRusso et al Annals of Oncology 2020). Based on these results, do the authors think this combination can be investigated in the setting of a clinical trial? Authors have not mentioned this in their discussion or conclusion.

Response: Thank you for your thoughtful comments. The abstract by LoRusso and colleagues [11] describes the lack of efficacy in adults with advanced solid tumors is indeed suggesting that the combination is not effective. However, this patient population was an unselected patient population. There are also publications suggesting some clinical responses in select patients with activation of the CDK4/6 and MAPK pathways[9]. In this cohort, 9 patients were treated and at least half of the patient had some clinical benefit. In addition, it is possible that DMG patients may tolerate higher doses of the combination resulting in more robust clinical benefit particularly if the combination is evaluated initially and not in recurrence. Therefore, we still believe that the combination is a promising combination for evaluation upfront in children with DMG, particularly in the subset of the tumors with MAPK pathway alterations such as PDGFRA signaling and CDK4/6 alterations with H3.3K27M as a biomarker for CDK4/6 alterations.

Page 29 , Line 637-641 : “With regards to clinical evaluation of the combination of CDK4/6 and MEK inhibition in patients with cancer, there was a phase Ib in unselected adult patients with solid tumors that did not observe any efficacy[11] while a case series of nine adult patients with solid tumor harboring CDK4/6 and MEK pathway alterations treated with CDK4/6 and MEK inhibitors noted some clinical benefit in approximately 50% of patients [9]. ”

3) Considering all the results, do the authors feel ribociclib is adding any value to trametinib's activity?

Response: We agree with the reviewer that trametinib is the more potent agent in this combination, potentially due to its better pharmacokinetics profile at least in part. However, as the combination significantly prolonged survival relative to trametinib monotherapy in the GEM, ribociclib is also an important contributor. In addition, this combination has shown preclinical efficacy across multiple types of tumors ranging from uveal melanoma[8], colorectal cancer[12], sarcomas[7, 10], and neuroblastomas[2] and now DMG.

Page 32-33, Line 715-726: “Future studies should also evaluate the safety profile of this combination therapy in tumor-bearing mice. Third, our data demonstrates that combination therapy with ribociclib and trametinib is effective against our GEM DMG model driven by PDGF signaling, H3.3K27M and p53 loss, and showing histologically high-grade H3.3K27M-positive tumors, but not the DMG PDX which harbors PDGFRA activating mutations, p53 LOF mutation, CDKN2A/B deletion, and MYC amplification. Future studies will elucidate the mechanisms for this discrepancy. One potential explanation is that MYC amplification may confer resistance to this combination as the study by Hart et al. noted that MYCN amplification conferred resistance to MEK inhibition[2]. Another potential explanation for the lack of efficacy in the PDX model is the testing in an immunodeficient model as multiple groups have noted greater efficacy of the combination in immunocompetent models and/or an immune mechanism as part of the anti-tumor effect of this combination[6, 7, 10].

References

1. Mayr L, Neyazi S, Schwark K, Trissal M, Beck A, Labelle J, et al. Effective targeting of PDGFRA-altered high-grade glioma with avapritinib. Cancer Cell. 2025;43(4):740-56.e8. Epub 20250313. doi: 10.1016/j.ccell.2025.02.018. PubMed PMID: 40086436.

2. Hart LS, Rader J, Raman P, Batra V, Russell MR, Tsang M, et al. Preclinical Therapeutic Synergy of MEK1/2 and CDK4/6 Inhibition in Neuroblastoma. Clin Cancer Res. 2017;23(7):1785-96. Epub 20161011. doi: 10.1158/1078-0432.

---

## [Editor Report · Decision Letter 1]

6 Nov 2025

Effects of Combination Therapy of a CDK4/6 and MEK Inhibitor in Diffuse Midline Glioma Preclinical Models

PONE-D-25-18253R1

Dear Dr. Becher,

We’re pleased to inform you that your manuscript has been judged scientifically suitable for publication and will be formally accepted for publication once it meets all outstanding technical requirements.

Kind regards,

Marielle Yohe

Academic Editor

PLOS ONE

Additional Editor Comments (optional):

Thank you so much for carefully addressing the concerns of the editor and reviewer 1. No further questions or comments.
---

## [Editor Report · Acceptance letter]

PONE-D-25-18253R1

PLOS One

Dear Dr. Becher,

I'm pleased to inform you that your manuscript has been deemed suitable for publication in PLOS One. Congratulations! Your manuscript is now being handed over to our production team.

Kind regards,

on behalf of

Dr. Marielle Yohe

Academic Editor

PLOS One